# RETHINKING CROSS-LINGUAL ALIGNMENT: BALANCING TRANSFER AND CULTURAL ERASURE IN MULTILINGUAL LLMS

## ABSTRACT

Cross-lingual alignment (CLA) aims to align multilingual representations, enabling Large Language Models (LLMs) to seamlessly transfer knowledge across languages. While intuitive, we hypothesize, this pursuit of representational convergence can inadvertently cause "cultural erasure"—the functional loss of providing culturally-situated responses that should diverge based on the query language. In this work, we systematically analyze this trade-off by introducing a holistic evaluation framework, the transfer-localization plane, which quantifies both desirable knowledge transfer and undesirable cultural erasure. Using this framework, we re-evaluate recent CLA approaches and find that they consistently improve factual transfer at the direct cost of cultural localization across all six languages studied. Our investigation into the internal representations of these models reveals a key insight: universal factual transfer and culturally-specific knowledge are optimally steerable at different model layers. Based on this finding, we propose Surgical Steering, a novel inference-time method that disentangles these two objectives. By applying targeted activation steering to distinct layers, our approach achieves a better balance between the two competing dimensions, effectively overcoming the limitations of current alignment techniques.

## 1 INTRODUCTION

Multilingual Large Language Models (LLMs) are expected to perform knowledge transfer uniformly across all languages (Li et al., 2024a; Lu & Koehn, 2025), transcending the inherent asymmetries in their training data (Ashrafimoghari, 2023). For example, a model that acquires knowledge in English for the question, *"What % of the body is water?"* should ensure that this knowledge is equally retrievable regardless of the query language. However, empirical studies have reported significant performance gaps across languages in multilingual tasks (Qi et al., 2023; Jiang et al., 2020; Kassner et al., 2021). To overcome these inconsistencies, multilingual LLMs rely on cross-lingual alignment, aiming at bringing different language representations closer together. Within this framing, inconsistencies across languages are typically regarded as undesirable (Jiang et al., 2020; Ohmer et al., 2023). However, this pursuit of uniformity creates a critical tension: what happens to knowledge that should be local? Consider the question (Figure 1): *"What is the emergency number?"* Does representational alignment cause the model to default to *"911"* regardless of the query language?

While prior work on cross-lingual alignment (CLA) has predominantly focused on its benefits for knowledge transfer, potential side effects remain underexplored. We address this gap by investigating a critical trade-off: the desirable transfer gained through alignment versus the undesirable loss of the model's ability to provide culturally localized responses. In doing so, we ask the following questions:

**How can we evaluate both the gains and losses of alignment?** We propose a holistic evaluation framework built on a two-dimensional transfer-localization plane (Section 3). The first axis measures desirable transfer, where a model should provide consistent responses across languages. The second axis measures cultural localization, the model's ability to tailor its responses to the cultural context inferred from the input language. Within this plane, we identify an undesirable quadrant where high transfer is achieved at the cost of cultural erasure—a regression in the model's ability to adapt.

Figure 1: Examples of intended convergence and desired divergence in outputs of multilingual LLMs. Universal questions (left) should result in a single, converged answer (knowledge transfer) regardless of the query languages, while culturally-specific questions (right) should result in divergent, localized answers (cultural localization) reflecting cultural context inferred from the input language.

**What hidden cultural costs accompany current cross-lingual alignment methods?** We reevaluate a series of popular CLA methods on the transfer-localization plane (Section 4) and show that while these methods improved knowledge transfer, they consistently degrade the model's ability to answer culturally specific questions, exposing a significant hidden cost.

**How can we design culturally-aware alignment techniques to better balance the trade-off?** By analyzing the model's internal representations, we identify a key distinction in how knowledge is encoded: while cross-lingual transfer is better realized within a model's middle layers, *cultural localization is predominantly encoded in the deeper layers*. Leveraging this insight, we introduce a simple, layer-specific intervention to steer the model towards both universal and local subspaces (Section 5). We show this method improves both transfer and localization across all CLA techniques, pushing performance into the desirable quadrant. Nevertheless, the trade-off is not fully eliminated, indicating that a residual loss of cultural nuance is inherent to the alignment process.

In summary, our work reframes the study of cross-lingual alignment by centering the critical trade-off between knowledge transfer and cultural localization, paving the way for the development of culturally-aware alignment in truly multilingual LLMs.

## 2 RELATED WORK

**The Root of Multilingual Gaps: Data and Representational Asymmetry** Performance gaps in multilingual models are often attributed to severe imbalances in their training data (Ashrafimoghari, 2023). This asymmetry leads to a model where knowledge is primarily encoded in the representations of high-resource languages (like English), which dominate pre-training corpora (Wenzek et al., 2020; Pfeiffer et al., 2022). Internally, this manifests as LLMs processing multilingual inputs by mapping them to a shared, language-agnostic semantic space—one that is often heavily biased towards English—before translating them back to the target language for the final output (Zhao et al., 2024; Wendler et al., 2024; Dumas et al., 2025). Consequently, the degree of alignment between English and non-English representations has become a reliable proxy for multilingual capability (Kargaran et al., 2025; Ravisankar et al., 2025), while performance degradation of non-English is often linked to failures in this internal convergence or translation process (Wang et al., 2025).

**Closing Multilingual Gaps: Cross-lingual Alignment** CLA approaches have introduced throughout the LLM development cicle. *During pre-training*, alignment is implicitly induced as a byproduct of training on parallel data, which act as cross-lingual representation anchors (Blum et al., 2025). *At post-training*, alignment is enhanced through multilingual instruction-tuning (Ouyang et al., 2022; Lai et al., 2023; Zhang et al., 2024), or by introducing objectives that explicitly encourage semantic alignment or language-agnostic retrieval (Lee et al., 2025; Liu & Niehues, 2025). *At inference time*, proposed CLA include steering representations towards English (Lim et al., 2025; Lu et al., 2025), merging task and language-specific adapters Zhao et al. (2025), swapping layers between specialized models Bandarkar et al. (2025), or simply translating queries into English externally (Banea et al., 2008; Etxaniz et al., 2024) or using cross-lingual thought prompting (Huang et al., 2023).

**Culturally-Situated LLMs: Desired Representation Localization** Cultural localization has become a central challenge for LLMs, with recent research establishing they exhibit a strong Western-centric bias (Bayramli et al., 2025; Zhou et al., 2025). In response, major research efforts focus on creating benchmarks to diagnose these biases, by curating multilingual datasets (Clark et al.,

2020; Salazar et al., 2025; Hasan et al., 2025); investigating social constructs through datasets on stereotypes (Bhutani et al., 2024); social norms (Forbes et al., 2020; Rao et al., 2025); or divergent cross-lingual perspectives on the same topics (Shwartz, 2022; Li et al., 2024b). Finally, recent work explores inference-time, culturally-aware approaches based on static methods (Arora et al., 2023; Lertvittayakumjorn et al., 2025; Li et al., 2024a) or adapted prompting with agents (Ki et al., 2025).

To date, research on **cross-lingual transfer and culturally-situated models has largely proceeded in isolation**, with the former focusing on enforcing cross-lingual representation alignment and the latter on localization rooted in cultural context. We unify these two strands with a framework (Section 3) designed to uncover the hidden costs of alignment (Section 4) and to develop interventions that balance shared knowledge with cultural specificity (Section 5).

## 3 Measuring the transfer-localization Trade-off

In this work, we propose a framework that measures both the benefits of cross-lingual representation alignment and the costs of losing cultural localization nuance during CLA. To formalize this, we introduce a typology of two distinct knowledge categories: *universal knowledge* refers to language-invariant knowledge, where a model's response should remain (semantically) consistent across languages. Conversely, *culturally-adaptive knowledge* is based on universal concepts but instantiated differently through local norms, cultural contexts, or regulations. In such scenarios, a model should preserve language-specific nuances, making output localization the intended behavior.

Building on this typology, we define two key metrics to evaluate CLA along a **transfer-localization** plane. As seen below, both metrics are defined as relative changes in performance compared to an unaligned baseline model, allowing us to precisely measure the impact of each alignment technique:

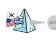 **Transfer**: We define transfer as the performance difference on *universal* knowledge tasks after applying an alignment method. It quantifies the *desirable* outcome of alignment: bridging the knowledge gap across different languages. A positive transfer score indicates that the model has successfully generalized knowledge from one language to another.

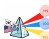 **Localization**: We define cultural localization as the performance difference on *culturally-adaptive* tasks. A negative score indicates cultural erasure, representing a functional loss in the model's ability to handle culturally specific questions.

To **operationalize** this framework, we employ benchmarks tailored to each knowledge type. We quantify knowledge transfer using Global MMLU (Singh et al., 2025, GMMLU), which contains universal multiple-choice questions across various academic and professional subjects. We measure cultural localization using a multilingual version of the BLEND benchmark (Myung et al., 2024), which is designed to evaluate knowledge of culturally and regionally specific concepts.[1] By plotting the change in GMMLU accuracy (Transfer) against the change in BLEND accuracy (Localization), we can map each CLA method to a point on the transfer-localization frontier, visualizing its trade-off.

## 4 Uncovering the Hidden Cost of Alignment

We revisit a series of recent CLA approaches and evaluate them under our proposed transfer-localization framework. First, we discuss preliminaries of the studied CLA techniques (§4.1), then describe our experimental setup (§4.2), and conclude with deep dives into the results (§4.3-§4.4).

### 4.1 Cross-Lingual Alignment Preliminaries

We focus on four recent CLA methods that have been proposed to foster representation alignment across languages. Some methods achieve this implicitly by training on parallel data, while others explicitly enforce alignment by directly manipulating or optimizing model representations, often

---

[1]To tailor BLEND to our needs, we automatically generate a decontextualized version by removing explicit localization context (e.g., "in Greece") from its questions (details in Appenfix A.2). This lets us test the model's ability to provide culturally-situated responses by inferring the right context from the language itself.

guiding them toward English latent subspaces. We consider a spectrum of approaches, covering two main paradigms: post-training and inference-time steering approaches which we detail below.

**Multilingual Instruction Tuning (MIST)**  employs a standard negative log likelihood (NLL) loss, which involves training on multilingual datasets of query-response pairs. The multilingual datasets are usually derived by extending English datasets through translation and training on such data is shown to enhance a model's generalization capabilities across various languages. In this case, representation alignment is implicitly enforced as a byproduct of training on parallel instruction tuning datasets (Lai et al., 2023; Blum et al., 2025).

**Middle-Layer Representation Alignment (MIDALIGN)**  introduces a more explicit alignment mechanism, alternating between a supervised fine-tuning (SFT) loss and a dedicated cross-lingual alignment loss (Liu & Niehues, 2025). Concretely, activations from the middle layer $\ell$ of the network are extracted for parallel texts ($\boldsymbol{h}_{\text{SRC}}^{\ell}$, $\boldsymbol{h}_{\text{TGT}}^{\ell}$) and mean-pooled over sequence. The alignment loss ($\mathcal{L}_{\text{MIDALIGN}}$, Eq. 1) is then formulated to maximize the similarity between translations, while minimizing the cosine similarity between non-translations within the same batch $\mathcal{B}$, which directly shapes the latent space to be more language agnostic. The loss is given as:

$$\mathcal{L}_{\text{MIDALIGN}} = -\log \frac{\exp\big(\cos(\boldsymbol{h}_{\text{SRC}}^{\ell}, \boldsymbol{h}_{\text{TGT}}^{\ell})\big)}{\sum_{\text{b}\in\mathcal{B}} \exp\big(\cos(\boldsymbol{h}_{\text{SRC}}^{\ell}, \boldsymbol{h}_{\text{b}}^{\ell})\big)}. \tag{1}$$

**Cross-lingual Optimization (CLO)**  aims at transferring an LLM's English capabilities to a target language by using a Cross-Lingual (CL) loss (Lee et al., 2025)—an adaptation of the Direct Preference Optimization objective (Rafailov et al., 2023). Concretely, for a non-English query $x_{\text{XX}}$, English responses $y_{\text{EN}}$ are suppressed, while in-language responses $y_{\text{XX}}$ are preferred, and vice versa; enabling the model to leverage its existing English knowledge for generating outputs in a target language. Formally, the loss is $\mathcal{L}_{\text{CLO}} = \lambda\,\mathcal{L}_{\text{SFT}} + (1-\lambda)\,\mathcal{L}_{\text{CL}}$ where $\mathcal{L}_{\text{SFT}}$ is applied on non-English query-response pair ($x_{\text{XX}}, y_{\text{XX}}$), and $\mathcal{L}_{\text{CL}}$ is given as follows:

$$\mathcal{L}_{\text{CL}} = -\mathbb{E}_{(x_{\text{EN}}, y_{\text{EN}}, y_{\text{XX}})\sim\mathcal{D}}[\log\sigma(z_{\text{EN}})] - \mathbb{E}_{(x_{\text{XX}}, y_{\text{XX}}, y_{\text{EN}})\sim\mathcal{D}}[\log\sigma(z_{\text{XX}})], \quad \text{where}$$

$$z_{\text{EN}} = \beta\left(\log\frac{\pi_\theta(y_{\text{EN}}|x_{\text{EN}})}{\pi_{\text{ref}}(y_{\text{EN}}|x_{\text{EN}})} - \log\frac{\pi_\theta(y_{\text{XX}}|x_{\text{EN}})}{\pi_{\text{ref}}(y_{\text{XX}}|x_{\text{EN}})}\right), \quad z_{\text{XX}} = \beta\left(\log\frac{\pi_\theta(y_{\text{XX}}|x_{\text{XX}})}{\pi_{\text{ref}}(y_{\text{XX}}|x_{\text{XX}})} - \log\frac{\pi_\theta(y_{\text{EN}}|x_{\text{XX}})}{\pi_{\text{ref}}(y_{\text{EN}}|x_{\text{XX}})}\right). \tag{2}$$

**English Steering (EN-steering)**  is an inference-time intervention based on contrastive activation addition (Rimsky et al., 2024), where "steering vectors" are computed to shift the model's distribution towards a desired behavior. In the context of CLA, Lim et al. (2025) propose to shift a model's latent space towards English motivated by prior work's observation that the shared latent space in multilingual LLMs is closer to English (Wendler et al., 2024). Following this, we sample contrastive pairs $\mathcal{S}$ consisting of English and non-English parallel queries ($x_{\text{EN}}$, $x_{\text{XX}}$). We then compute the average differences between the activations $\boldsymbol{h}^{\ell}(x)$ at layer $\ell$ over all pairs, resulting in an English steering vector, $\boldsymbol{v}_{\text{EN}}^{\ell}$. During inference, this vector is then scaled by a factor $\gamma$ and added to $\boldsymbol{h}^{\ell}(x)$ to produce the modified activation, $\tilde{\boldsymbol{h}}^{\ell}(x)$ as shown below:

$$\boldsymbol{v}_{\text{EN}}^{\ell} = \frac{1}{|\mathcal{S}|}\sum_{x_{\text{EN}}, x_{\text{XX}}\in\mathcal{S}}\big(\boldsymbol{h}^{\ell}(x_{\text{EN}}) - \boldsymbol{h}^{\ell}(x_{\text{XX}})\big), \quad \tilde{\boldsymbol{h}}^{\ell}(x) = \boldsymbol{h}^{\ell}(x) + \gamma\,\boldsymbol{v}_{\text{EN}}^{\ell}. \tag{3}$$

## 4.2 Experimental Settings

**Evaluation Details**  We measure the transfer-localization trade-off with multiple-choice datasets: GMMLU and BLEND, on six languages: Spanish (ES), Indonesian (ID), Korean (KO), Greek (EL), Chinese (ZH), and Arabic (AR).[2] As BLEND does not include a development set, we create one by setting aside 200 randomly chosen samples from the test data. The remaining data is what constitutes our true test set. We further split the development set into two: 100 samples for steering vector extraction and 100 for layer-wise analysis.[3] The model's accuracy is determined by computing the log likelihood of each answer option and selecting the one with the highest probability.

---

[2]For languages that are associated with multiple regions in BLEND, we choose Spain for Spanish and South Korea for Korean.

[3]Detailed statistics of the two benchmarks are provided in the Appendix A.2.

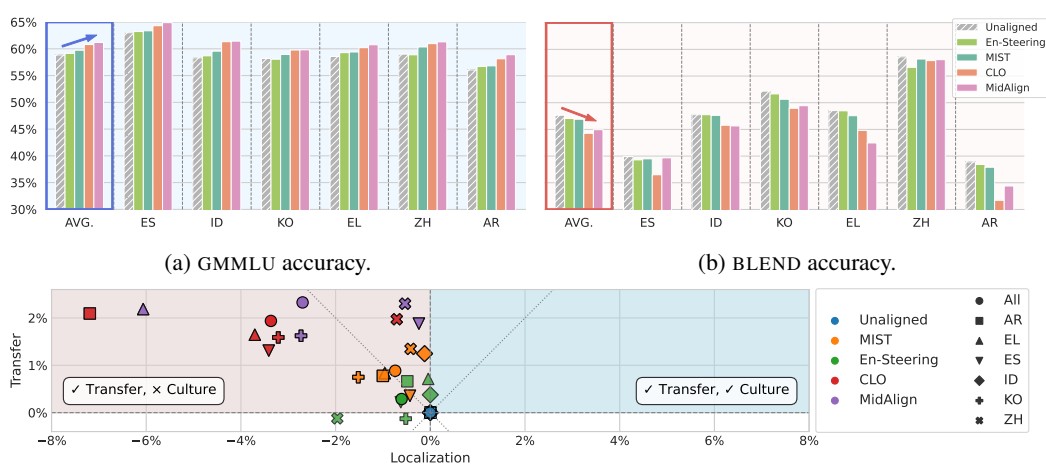

(a) GMMLU accuracy.

(b) BLEND accuracy.

(c) Transfer-Localization trade-offs ($\Delta = \text{Acc}_{\text{CLA}} - \text{Acc}_{\text{UNALIGNED}}$) on GMMLU and BLEND.

Figure 2: Competing results of CLA approaches on knowledge transfer and cultural localization. **Improvements in CLA come at a *consistent* cost of cultural localization across all languages.**

**Model Training** We detail the training settings for each CLA approach below:

1. MIST: We use 6K English instruction–response pairs (first-turn only) from the OpenAssistant dataset (Köpf et al., 2023) and translate them into all six languages with Google Translate.[4] We refer to this dataset as OpenAssistantXX and use it to train MIST.

2. MIDALIGN: We use OpenAssistantXX and FLORES (NLLB Team, 2024), alternating between SFT and MIDALIGN loss, respectively. Layer 24 is set as the middle layer for extracting representations to compute the alignment loss, following prior work.

3. CLO: We create preference pairs using the multi-way parallel OpenAssistantXX dataset and use $\lambda = 0.5$ and $\beta = 1.0$, following Lee et al. (2025).

4. EN-steering: While activation steering can be applied to any model, we default to the UNALIGNED model unless noted. Activations are extracted using 100 samples from GMMLU's dev set. We set $\gamma$ to 2. We use layer-wise Principal Component Analysis (Wold et al., 1987, PCA) analysis and identify layers 16-32 to exhibit the highest overlap of hidden activations across languages which is necessary for steering to be effective. We apply EN-steering at layer 20 based on the accuracy on the development set.

**Model Architecture** We conduct all our experiments using the Gemma3 12B pre-trained model (Gemma Team, 2025), which consists of 48 transformer layers. Gemma3 models have been trained on a vast amount of multilingual datasets, natively supporting 35+ languages, which makes it a good candidate to evaluate cross-lingual transfer. All post-training models (MIST, CLO, MIDALIGN) are trained on seven languages including English, for only one epoch, updating all parameters.[5]

### 4.3 MAPPING CROSS-LINGUAL ALIGNMENT TO THE TRANSFER-LOCALIZATION PLANE

Figure 2 presents performance of the pre-trained model (i.e., UNALIGNED) and each of the CLA approaches on GMMLU (universal knowledge) and BLEND (culturally-adaptive knowledge) datasets.

**CLA effectively improves knowledge transfer.** When evaluating universal knowledge transfer on GMMLU (Fig. 2a), we observe that all CLA approaches generally improve performance over the UNALIGNED baseline, across six non-English languages. However, the magnitude of this improvement varies. For instance, methods like MIDALIGN (+2.3%) and CLO (+1.9%) consistently deliver the largest gains, suggesting that more explicit alignment is highly effective at bridging significant

---

[4]https://translate.google.com

[5]More details about post-training are in Appendix A.1.

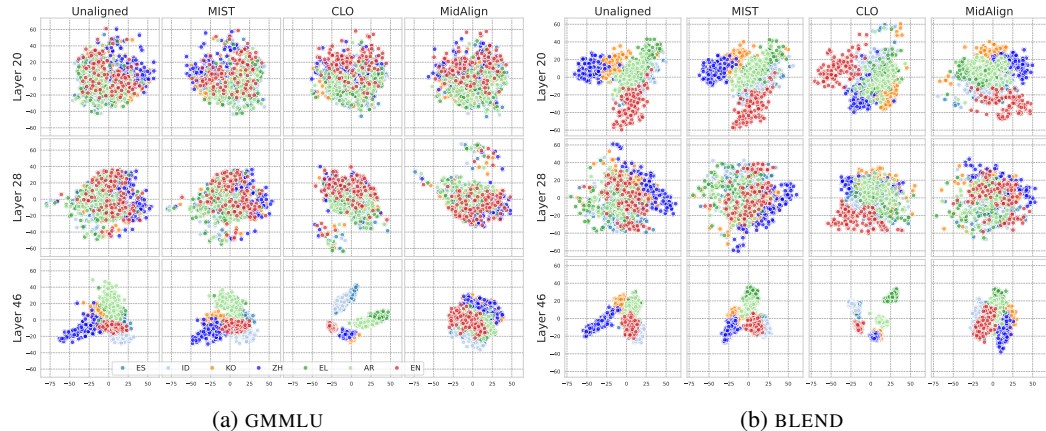

(a) GMMLU                  (b) BLEND

Figure 3: PCA projections of hidden representations across UNALIGNED and CLA methods. As CLA methods are applied, languages cluster more tightly, signaling stronger convergence. Yet, convergence differs by the nature of the datasets: GMMLU merges starting in the middle layers, whereas BLEND maintains separation until later stages, persisting even after CLA.

knowledge gaps. In contrast, MIST (+0.9%) and EN-steering (+0.3%) provide more modest, though still positive, gains. This finding is in complete alignment with results from prior work and indicates that when universal transfer is the target, all studied approaches are deemed successful.

**CLA results in potential cultural erasure as suggested by the accuracy drop in BLEND.** Results on the culturally adaptive BLEND dataset (Fig. 2b) reveal the significant cost of this alignment. All alignment methods lead to a degradation in performance on culturally specific questions. This loss of nuance is particularly pronounced for the most effective transfer methods. For example, CLO, which showed strong gains on GMMLU, consistently causes the most substantial performance drop (-3.4%) on BLEND across nearly all languages. This suggests that its aggressive representation alignment overwrites culturally specific information. Conversely, MIST, which enforces alignment more implicitly, induces the least amount of cultural erasure, preserving cultural knowledge more effectively than other methods.

**CLA exhibits transfer-localization tradeoffs.** To better show these competing outcomes, we plot the performance of each alignment method and language on a transfer-localization plane (Fig 2c). This plot positions each model-language pair based on its transfer gain (GMMLU improvement, y-axis) against its cultural localization (BLEND performance change, x-axis). The resulting frontier clearly illustrates the trade-off: methods that push further up (gaining transfer) invariably push further to the left (incurring erasure). A closer look at the frontier reveals distinct behaviors. The most aggressive alignment methods, MIDALIGN (purple) and CLO (red), occupy the top-left region of the plot, while "safer" approaches with minimal cultural erasure constitute less powerful options for generalization, with certain languages such as Korean and Chinese even exhibiting degradation in transfer. Finally, this plot highlights that for many CLA methods, the cost of erasure outweighs the benefit of transfer. This establishes the central challenge for our next section: how to move beyond this frontier and achieve transfer without hurting cultural localization.

## 4.4 THE INTERNAL DYNAMICS OF CROSS-LINGUAL ALIGNMENT

How do CLA approaches alter the model's internal representation space? We analyzed PCA projections of hidden states from various layers (middle: 20, deep: 28, outer: 47). As shown in Figure 3, we compared the unaligned base model to three progressively aligned models—MIST, CLO, and MIDALIGN—across the GMMLU and BLEND datasets.[6]

Our analysis reveals two key dynamics. First, the alignment process differs significantly depending on the nature of the data. On the universal GMMLU dataset, CLA methods successfully merge

---

[6]A comprehensive set of plots for all layers can be found in the Appendix B.2.

representations in the middle layers as intended. However, on the cultural BLEND dataset, language representations remain largely separable in these same middle layers, with alignment only beginning to emerge deeper in the model. Surprisingly, this cultural separability persists even in the most stringently aligned models. Second, regardless of the dataset, CLA methods (especially CLO and MIDALIGN) induce a stronger representational convergence in the deeper layers (e.g., layer 28). This raises the question: given the persistent representational differences on cultural data, could the associated performance losses on BLEND be recovered using techniques like representation steering?

## 5 BALANCING TRANSFER AND CULTURAL ERASURE

If CLA suppresses a model's ability to use language as a cultural cue, is that knowledge permanently erased or merely inaccessible? We start by exploring how existing activation steering techniques can be used to probe for localized knowledge (§5.1). Then, we present a crucial finding from our representation analysis: knowledge transfer and cultural localization are optimally steered at different layers (§5.2). Finally, we use this insight to better balance transfer and localization for CLA (§5.3).

### 5.1 PROBING FOR LOCALIZED KNOWLEDGE WITH ACTIVATION STEERING

Table 1: Transfer-Localization trade-offs for different steerings methods (applied on middle layer; avg. across langs).

| CLA | GMMLU (%) | BLEND (%) |
|---|---|---|
| MIST | 59.74 | 46.90 |
| + EN-steering | 59.90 ↑ 0.16 | 46.45 ↓ 0.45 |
| + LOC-steering | 59.60 ↓ 0.14 | 48.12 ↑ 1.22 |

To investigate the extent to which localized knowledge remains accessible within aligned models, we adopt the localized activation steering method of Veselovsky et al. (2025). Concretely, we use pairs of inputs with and without cultural context $(x_{\text{CON}}, x_{\text{DECON}})$ to derive a localizing vector $\boldsymbol{v}_{\text{LOC}}^{\ell}$ (LOC-steering), pushing the model toward local subspaces.[7]

$$\boldsymbol{v}_{\text{LOC}}^{\ell} = \frac{1}{|\mathcal{S}'|} \sum_{x_{\text{CON}}, x_{\text{DECON}} \in \mathcal{S}'} \left( \boldsymbol{h}^{\ell}(x_{\text{CON}}) - \boldsymbol{h}^{\ell}(x_{\text{DECON}}) \right), \quad \tilde{\boldsymbol{h}}^{\ell}(x) = \boldsymbol{h}^{\ell}(x) + \gamma \, \boldsymbol{v}_{\text{LOC}}^{\ell}. \quad (4)$$

As shown in Table 1, applying LOC-steering instead of EN-steering at the MIST's middle layer improves its ability to provide culturally situated responses. This indicates that cultural knowledge is not permanently erased but—at least to an extent— suppressed, capable of being reactivated through targeted steering. At the same time, this improvement on the cultural localization axis comes at a cost: universal transfer on GMMLU degrades $0.3\%$ from EN-steering, **suggesting that transfer and cultural localization are not optimally co-located within the same model layers**.

### 5.2 KNOWLEDGE TRANSFER AND CULTURAL LOCALIZATION PEAK AT DIFFERENT LAYERS

The above observations naturally prompt us to ask: where within a model's layers are cross-lingual transfer and cultural localization most effectively realized? We analyze the angle between the EN-steering and LOC-steering vectors to identify layers where localization and transfer are disentangled. Intuitively, for these interventions to operate independently, their vectors should be orthogonal. As shown in Figure 4a, this condition is met in the model's deeper layers, which approach orthogonality, but not in shallower layers where the vectors are closely aligned. Therefore, deeper layers (peaking at 28) are optimal for applying localization steering with minimal interference, while shallower layers (e.g., 20) risk conflicting signals.[8]

To systematically validate our observation, we analyze the layer-wise effects of applying EN-steering and LOC-steering in isolation on a held-out dev set.[9] The results, shown in Figure 4b, confirm our hypothesis from the angular analysis: while universal transfer (EN-steering) is most effective in the middle layers (peaking at layer 20), cultural localization (LOC-steering) performs optimally in the

---

[7]The localized steering vectors are extracted using the BLEND development set described in Section 4.2.

[8]More details of this analysis in Appendix B.1.

[9]This is the second development set, which is distinct from the one used to extract steering vectors. Test set result is available in Appendix Figure 9, showing similar trend.

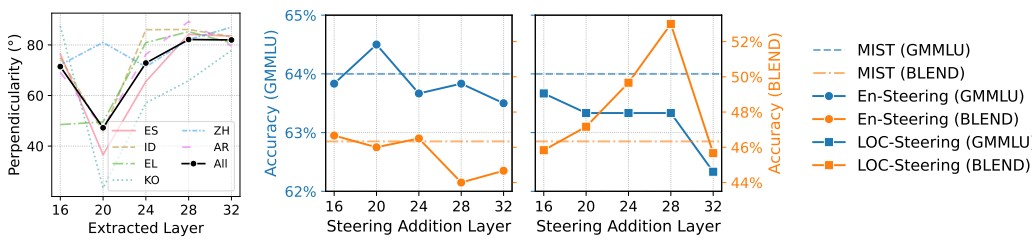

(a) Perpendicularity.  (b) Layer-wise performance on GMMLU and BLEND.

Figure 4: Layer-wise analysis of EN- and LOC-steering on MIST for GMMLU and BLEND dev set (right) and perpendicularity between two kinds of vectors (left). **Cultural localization is optimally located in deeper layers, where the EN- and LOC vectors are also most orthogonal to each other.**

deeper layers (peaking at layer 28). This finding echoes the hidden representation projection results in (§4.4), where middle-layer representations across languages in BLEND stay separated, making it unsuitable for effective steering. This layered separation has a critical practical implication: applying LOC-steering at a deeper layer (e.g., 28) significantly boosts performance on culturally situated questions without degrading universal transfer performance. Accordingly, we apply LOC-steering at a deeper layer by default in subsequent experiments.

## 5.3 PUSHING THE TRANSFER-LOCALIZATION FRONTIER WITH SURGICAL STEERING

Motivated by our analysis above, we propose Surgical Steering (SUR-steering): applying the EN-steering vector at an earlier layer $\ell_{\text{EN}}$ and the LOC-steering vector at a deeper layer $\ell_{\text{LOC}}$ to have more controlled alignment.[10] Formally, our surgical intervention is defined as follows:

$$\tilde{h}_l(x) = h_l(x) + \gamma\, \mathbf{1}_{l=\ell_{\text{EN}}}\, v^l_{\text{EN}} + \gamma\, \mathbf{1}_{l=\ell_{\text{LOC}}}\, v^l_{\text{LOC}}, \quad l \in \{1, \dots, |L|\}. \tag{5}$$

**Superiority of SUR-steering.**  Figure 5a shows that SUR-steering achieves a more favorable trade-off than applying either EN-steering or LOC-steering alone. It surpasses EN-steering in cross-lingual transfer while simultaneously improving cultural localization. This demonstrates that combining steering at distinct layers provides finer control over the alignment process, pushing the Pareto frontier towards a more optimal state. Language-wise results are in Appendix Table 3.

**General Steerability of CLA Models.**  Our experiments also reveal a broader insight: all tested CLA approaches remain steerable. As shown by the orange and green stars in Figure 5a, applying SUR-steering to models already trained with MIDALIGN and CLO yields further improvements in both transfer and localization. The gain for MIDALIGN is smaller, however, suggesting a saturation effect in models that already possess high transfer capabilities.

**Addressing English Bias with SUR-steering.**  Given the hypothesis that multilingual LLMs improve cross-lingual transfer by aligning representations towards English (Wendler et al., 2024), we assess whether surgical steering improves cultural localization by suppressing the English-centric responses. We extract all the queries from BLEND (∼40%) that include an answer associated with English-speaking countries (US/UK) and measure the proportion of times the model selects this option as an answer for a non-English query. Figure 5b shows that first, as the training progress, the model's tendency to select the English option increases across all CLA methods (MIST, CLO, MIDALIGN), validating this hypothesis.[11] However, applying SUR-steering to all approaches reduces this bias significantly (up to 4%), showing its effectiveness in steering the model away from English-centric responses. We note that the bias, however, is not completely removed as even when steering the UNALIGNED model, the accuracy of English responses remains high at 30.2%.

---

[10]Specifically, we simultaneously apply EN-steering on layer 20, and LOC-steering on layer 28 for the models based on Gemma3 12B, which has $|L| = 48$ layers.

[11]We also show the accuracy trends for all CLA approaches in the Appendix Figure 15.

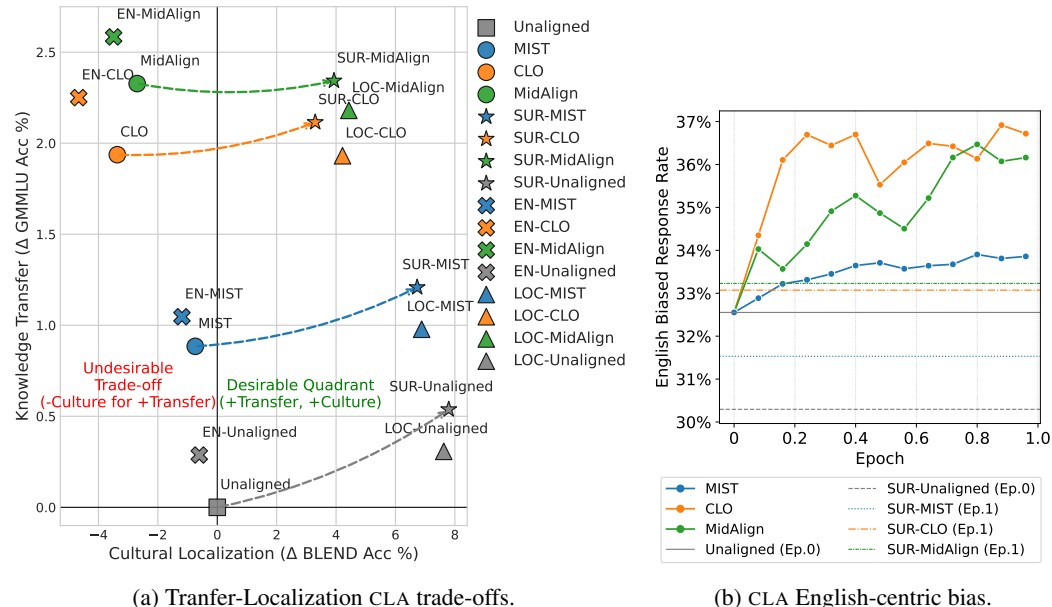

(a) Tranfer-Localization CLA trade-offs.

(b) CLA English-centric bias.

Figure 5: Left: Trade-offs between transfer and localization with steering methods. EN-steering, LOC-steering, and SUR-steering are applied on top of unalinged (square) and different post-training methods (circles), indicated by the same color or by the connecting gray dotted line. Right: Tracking English-bias of post-training CLA methods and the impact of SUR-steering on all approaches.

**The Irrecoverable Trade-off.** Critically, the fundamental trade-off persists. Models with stronger alignment, such as MIDALIGN and CLO, are less responsive to steering than the UNALIGNED or MIST models (Fig. 5a). This indicates that while cultural knowledge is partially recoverable, some cultural nuances are irrevocably lost during the alignment process. The same applies to English-biased responses (Fig. 5b): although steering alleviates the bias to some extent, the post-trained CLA models never fully return to the point of the UNALIGNED baseline, indicating inherent limits to what the steering can recover.

## 6 CONCLUSION

In this work, we address the critical trade-off between knowledge transfer and cultural localization in cross-lingual alignment. We introduce a holistic framework to systematically measure this trade-off, quantifying not only the gains in universal knowledge transfer but also the loss of cultural specificity. Our empirical analysis confirms that existing alignment methods consistently improve transfer at the direct cost of cultural localization. To mitigate this, we propose a simple yet effective method using controllable activation steering. We demonstrate that by disentangling universal and localized steering at different, optimal layers, we can improve performance on culturally situated tasks without compromising transfer. This reveals a key insight: cultural knowledge is not permanently erased by alignment but is instead suppressed, making it partially recoverable through targeted, layer-specific interventions. Nevertheless, our findings suggest that some cultural nuances are irretrievably lost, highlighting a persistent and perhaps unavoidable cost of the cross-lingual alignment process.

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

## THE USE OF LARGE LANGUAGE MODELS

We use LLM to partially refine or polish writing at the sentence level (e.g., fixing grammar, rewording sentences)

# A  ADDITIONAL EXPERIMENTAL DETAILS

## A.1  TRAINING DETAILS

We use 16 TPUv4 for post-training and set the batch size to 16 for all experiments. We use a maximum sequence length of 640 tokens. The peak learning rate is set to 5e-5 for MIST and 5e-4 for the rest. All our implementations are based on `Flax` (Heek et al., 2024), a neural network library for `jax` (Bradbury et al., 2018).

## A.2  BENCHMARKS AND EVALUATION

For creating decontextualized BLEND queries, we prompt Gemini 2.5 Flash with the instruction provided in Figure 7. Outputs are automatically checked against the originals, and cases with excessive reduction or unintended content changes are filtered and re-processed with Gemini 2.5 Pro. Since BLEND has English multiple choice options, we translate the non-English choices provided in the dataset with Gemini 2.5 Flash using the prompt shown in Figure 8. For cases where the translation output does not match the predefined format, we re-translate using Gemini 2.5 Pro.

The prompt used for the evaluation of multiple choice questions is shown in Figure 6.

For the development set, we randomly select 200 samples: from the development split of GMMLU and from the original set of BLEND. To extract the EN-steering and LOC-steering vectors, we use 100 samples (Dev1), reserving the remaining 100 for layer-wise analysis to determine the optimal layer (Dev2). Detailed statistics for both benchmarks are shown in Table 2.

Table 2: GMMLU and BLEND Statistics

| | | GMMLU | | | BLEND | | | | |
|---|---|---|---|---|---|---|---|---|---|
| Code | Language | Dev1 | Dev2 | Test | Region | Extracted | Dev1 | Dev2 | Test |
| ES | Spanish | 100 | 100 | 14042 | Spain | 19280 | 100 | 100 | 19080 |
| ID | Indonesian | 100 | 100 | 14042 | Indonesia | 18417 | 100 | 100 | 18217 |
| KO | Korean | 100 | 100 | 14042 | South Korea | 21439 | 100 | 100 | 21239 |
| EL | Greek | 100 | 100 | 14042 | Greece | 20383 | 100 | 100 | 20183 |
| ZH | Chinese | 100 | 100 | 14042 | China | 20410 | 100 | 100 | 20210 |
| AR | Arabic | 100 | 100 | 14042 | Algeria | 20364 | 100 | 100 | 20164 |

```
{question}
Without any explanation, choose only one from the given alphabet choices (e.g., A, B, C, D).
A. {option_a}
B. {option_b}
C. {option_c}
D. {option_d}
Answer:
```

Figure 6: The prompt template used for our multiple-choice question experiments.

```
Analyze the following list of question. Identify the shared, core question by removing the
    specific location (e.g., "in US", "in UK", "in West Java") from the end of each sentence.
- Remove the specific context to create a natural, generalized question.
- Do not put [country] in it just remove the country name and make it natural.
- Do not paraphrase the original input. Just try to minimally remove the context from the
    input.
Provide only this single, decontextualized question as the output.
Input: {Question}
Output:
```

Figure 7: The prompt template used for decontextualizing a query.

```
You are a professional translator, translating from English to {language} spoken in {
    country_name}.
Translate the given list of English keywords e.g. [key1, key2,...] and output in a dictionary
    format e.g. {{key1: translation1, key2: translation2, ...}}.
- A key that represents numerical data, a date, or a time (e.g., "123", "1,000", "10:30",
    "12/25") MUST be copied to its value instead of being translated.
- All other keys should be translated.
- All translation values MUST be a single string.
- If a hint is provided below for a specific keyword, you MUST use one of the suggested
    translations.
{hint_phrase}
List of keywords: {options}
Do NOT include any explanatory text, comments, or markdown formatting.
```

Figure 8: The prompt template used for translating options.

# B  ADDITIONAL ANALYSIS

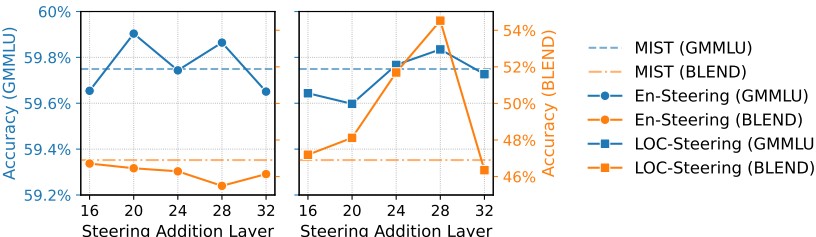

Figure 9: Layer-wise performance of EN-steering and LOC-steering of MIST on GMMLU and BLEND test set. We observe that cultural steering is optimally located at deeper layers. The trends are similar in the development set (Figure 4b).

## B.1  PERPENDICULARITY ANALYSIS BETWEEN EN-STEERING AND LOC-STEERING

We quantify perpendicularity as the deviation of the inter-vector angle from orthogonality between EN-steering vector and LOC-steering extracted from each layer. More specifically, we calculate a perpendicularity score $S_{\text{PER}}^\ell$ between $\boldsymbol{v}_{\text{EN}}^\ell$ and $\boldsymbol{v}_{\text{LOC}}^\ell$ at layer $\ell$ based on the closeness of the vector angle to 90 degrees as follows:

$$S_{\text{PER}}^\ell = 90 - \left| \left( \frac{180}{\pi} \arccos \left( \frac{\boldsymbol{v}_{\text{LOC}}^\ell \cdot \boldsymbol{v}_{\text{EN}}^\ell}{\|\boldsymbol{v}_{\text{LOC}}^\ell\|\|\boldsymbol{v}_{\text{EN}}^\ell\|} \right) \right) - 90 \right|. \quad (6)$$

A score of 90 means the vectors are perfectly perpendicular (90°), and a score of 0 means the vectors are perfectly parallel (0° or 180°). Overall, shallower layers exhibit lower perpendicularity between EN-steering and LOC-steering, whereas deeper layers approach closer to orthogonality (Figure 10). This implies that additional localization interventions can operate with minimal interference in deeper layers (peaking at 28), in contrast to shallower layers where the effects of transfer and localization are more entangled (as seen in layer 20).

## B.2  LAYER-WISE PCA ANALYSIS ACROSS CROSS-LINGUAL ALIGNMENT

Following Lim et al. (2025), we conduct a layer-wise PCA analysis with the extracted activations. In Figure 11, each color represents a different language activation extracted from the GMMLU samples. Unlike the early or late layers, where languages appear more easily separated, the middle layers (16–32) show better overlaps across languages. We focus our analysis on these layers, where the representations become more tightly clustered, suggesting that steering vectors are most effective in this region. A PCA plot for BLEND is in Figure 12.

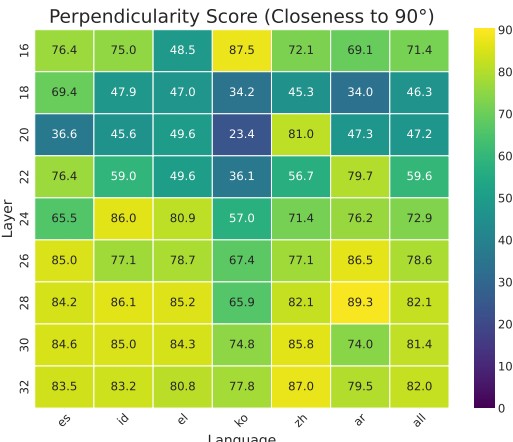

Figure 10: Layer-wise perpendicularity analysis between EN-steering and LOC-steering

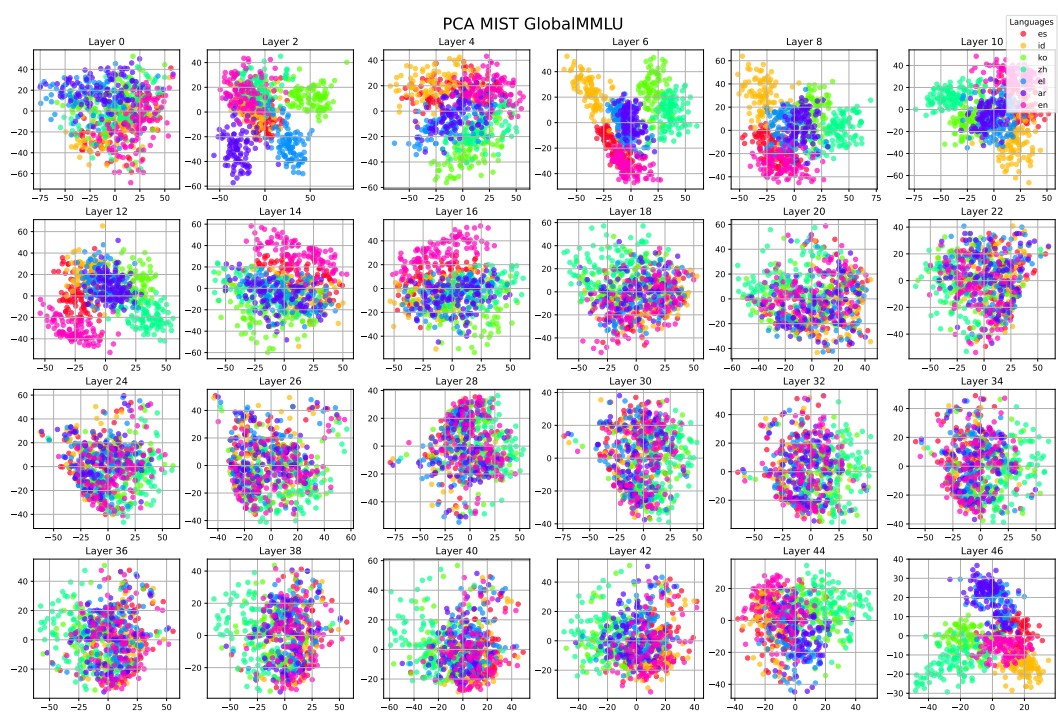

Figure 11: Layer-wise PCA plots of MIST with activations from GMMLU samples. Each color represents a different language.

### B.3 LAYER-WISE PCA ANALYSIS ON ACTIVATION STEERING

Next, we examine how activation steering alters the geometry of hidden representations across layers. Figure 13 shows PCA projections of Spanish and English representations from the MIST model (Orig, circle), along with their transformations under EN-steering (cross) and LOC-steering (diamond) interventions. By definition, EN-steering shifts representations toward English, consistently pushing Spanish vectors closer to the English cluster. In the earlier layers, the directions of the original English and localized representations often exhibit parallel or counter-parallel tendencies. However, starting around layer 26 and becoming especially evident at layers 28 and 30, the vectors

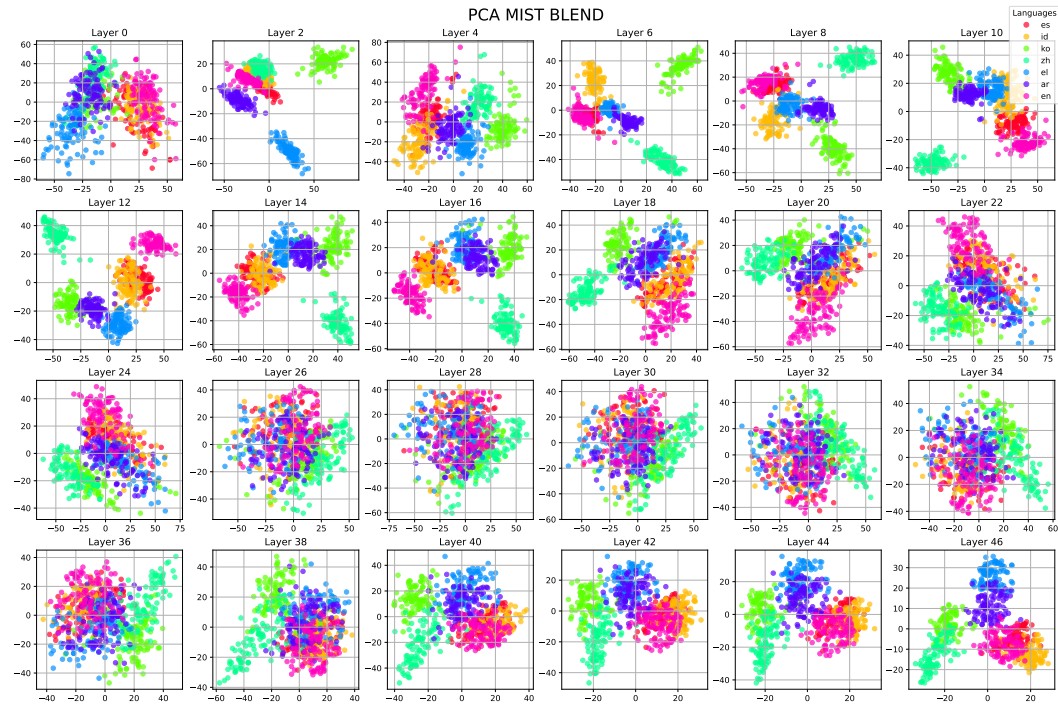

Figure 12: Layer-wise PCA plots of MIST with activations from BLEND samples. Each color represents a different language.

reveal a near-perpendicular relationship. Even in this 2D projection, this perpendicularity is clearly observable, consistent with our earlier findings in Appendix B.1.

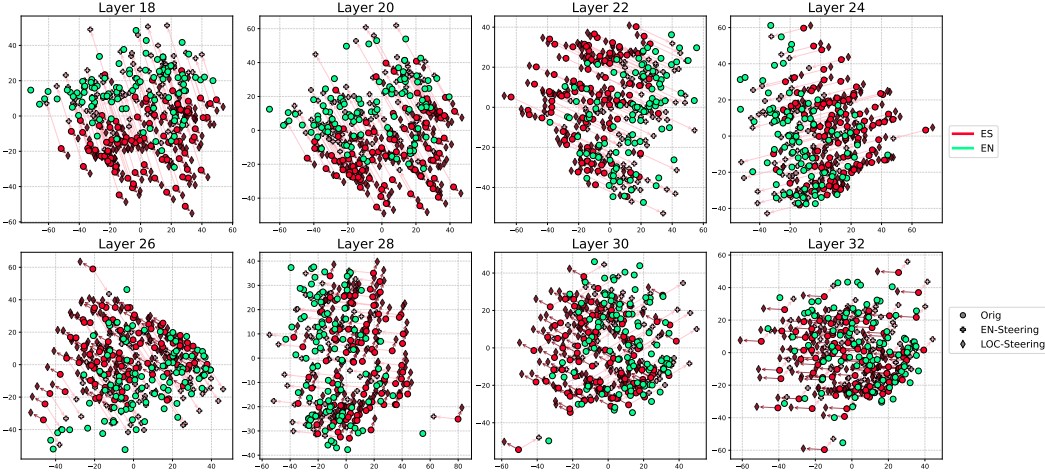

Figure 13: Layer-wise PCA plots of MIST. Each color represents a different language. PCA plots of Spanish and English hidden representations from MIST and its steering vectors. By definition, EN-steering shifts Spanish vectors toward the English cluster, while LOC-steering directs them toward localized subspaces.

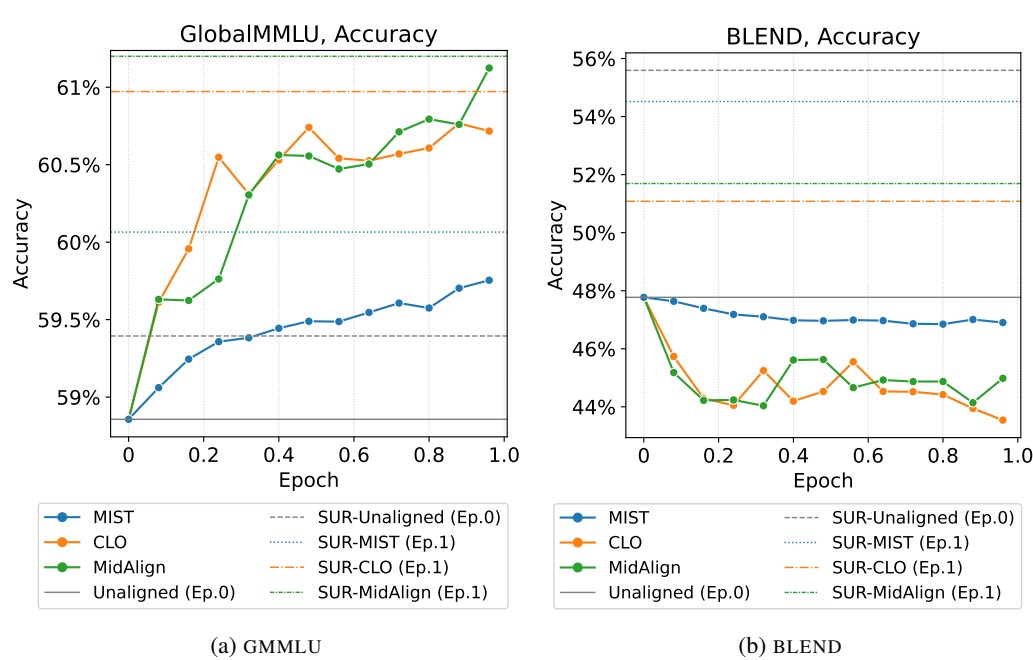

(a) GMMLU

(b) BLEND

Figure 14: Learning dynamics of post-training CLA methods across alignment epochs on GMMLU and BLEND.

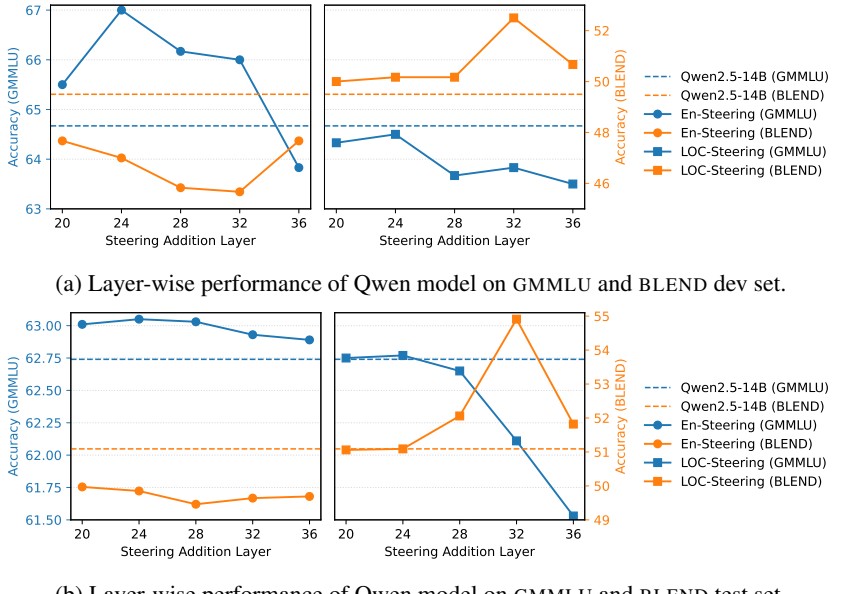

(a) Layer-wise performance of Qwen model on GMMLU and BLEND dev set.

(b) Layer-wise performance of Qwen model on GMMLU and BLEND test set.

Figure 15: Layer-wise performance of Qwen model (UNALIGNED) on GMMLU and BLEND.

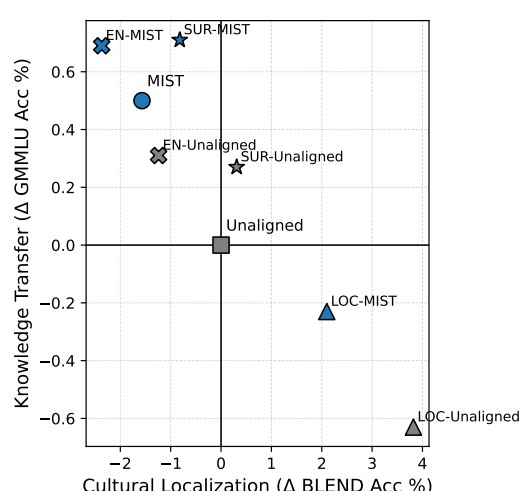

Figure 16: Transfer-Localization CLA trade-offs of Qwen (UNALIGNED and MIST) model.

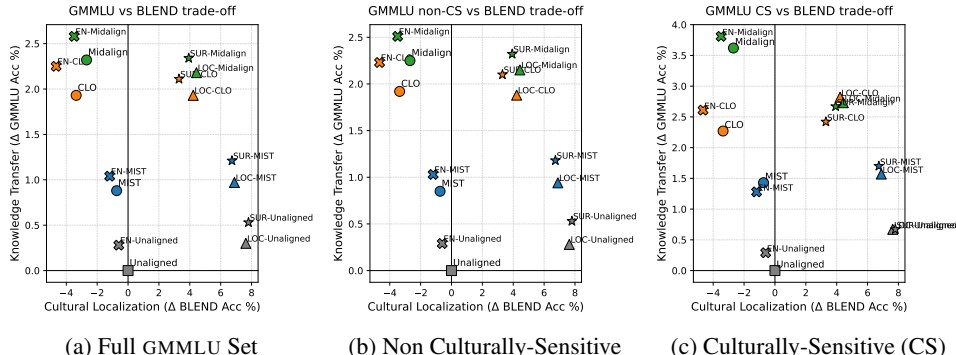

(a) Full GMMLU Set    (b) Non Culturally-Sensitive    (c) Culturally-Sensitive (CS)

Figure 17: Tranfer-Localization CLA trade-offs on different subset of GMMLU.

Table 3: Accuracy (%) of all CLA methods on GMMLU and BLEND test set. * denotes steering applied on a middle layer (20) and ** on a deeper layer (28). SUR-steering simultaneously apply EN-steering on the middle layer and LOC-steering on the deeper layer.

| GMMLU (%) | All | ES | ID | KO | EL | ZH | AR |
|---|---|---|---|---|---|---|---|
| UNALIGNED | 58.86 | 63.01 | 58.32 | 58.19 | 58.57 | 59.00 | 56.05 |
| + EN-steering * | 59.14 | 63.23 | 58.70 | 58.06 | 59.29 | 58.88 | 56.71 |
| + LOC-steering * | 58.87 | 62.87 | 58.28 | 58.18 | 58.47 | 59.32 | 56.12 |
| + LOC-steering ** | 59.16 | 63.28 | 58.94 | 58.14 | 58.39 | 59.69 | 56.55 |
| + SUR-steering | 59.39 | 63.45 | 59.15 | 58.50 | 59.22 | 59.27 | 56.79 |
| MIST | 59.74 | 63.37 | 59.56 | 58.94 | 59.41 | 60.35 | 56.82 |
| + EN-steering * | 59.90 | 63.75 | 59.69 | 58.79 | 59.87 | 60.03 | 57.29 |
| + LOC-steering * | 59.60 | 63.30 | 59.28 | 58.83 | 59.41 | 60.16 | 56.60 |
| + LOC-steering ** | 59.83 | 63.65 | 59.88 | 58.48 | 59.18 | 60.55 | 57.26 |
| + SUR-steering | 60.07 | 64.01 | 60.15 | 58.99 | 59.79 | 60.09 | 57.36 |
| CLO | 60.79 | 64.32 | 61.33 | 59.78 | 60.22 | 60.97 | 58.14 |
| + EN-steering * | 61.11 | 64.93 | 61.63 | 60.08 | 60.76 | 60.77 | 58.46 |
| + LOC-steering * | 60.79 | 64.48 | 61.29 | 59.72 | 60.40 | 60.93 | 57.91 |
| + LOC-steering ** | 60.79 | 64.00 | 61.16 | 59.93 | 60.52 | 60.71 | 58.40 |
| + SUR-steering | 60.97 | 64.44 | 61.28 | 59.91 | 60.74 | 60.68 | 58.79 |
| MIDALIGN | 61.18 | 64.89 | 61.44 | 59.81 | 60.76 | 61.30 | 58.90 |
| + EN-steering * | 61.44 | 65.07 | 61.69 | 60.22 | 61.22 | 61.20 | 59.24 |
| + LOC-steering * | 61.22 | 64.96 | 61.45 | 60.04 | 60.78 | 61.23 | 58.86 |
| + LOC-steering ** | 61.04 | 64.44 | 61.38 | 59.46 | 60.33 | 61.64 | 58.97 |
| + SUR-steering | 61.20 | 64.83 | 61.39 | 59.55 | 60.48 | 61.62 | 59.33 |

| BLEND (%) | All | ES | ID | KO | EL | ZH | AR |
|---|---|---|---|---|---|---|---|
| UNALIGNED | 47.64 | 39.91 | 47.75 | 52.17 | 48.53 | 58.60 | 38.91 |
| + EN-steering * | 47.04 | 39.29 | 47.75 | 51.65 | 48.48 | 56.64 | 38.42 |
| + LOC-steering * | 48.71 | 40.39 | 48.76 | 54.51 | 49.43 | 60.02 | 39.13 |
| + LOC-steering ** | 55.27 | 48.97 | 54.25 | 64.90 | 57.10 | 65.14 | 41.27 |
| + SUR-steering | 55.44 | 48.60 | 55.30 | 65.07 | 57.77 | 65.24 | 40.65 |
| MIST | 46.90 | 39.48 | 47.63 | 50.65 | 47.56 | 58.18 | 37.90 |
| + EN-steering * | 46.45 | 39.33 | 47.92 | 50.24 | 47.28 | 56.62 | 37.35 |
| + LOC-steering * | 48.12 | 39.82 | 48.44 | 53.32 | 48.60 | 60.07 | 38.46 |
| + LOC-steering ** | 54.53 | 48.19 | 53.89 | 63.59 | 56.67 | 64.46 | 40.36 |
| + SUR-steering | 54.37 | 47.56 | 54.48 | 63.20 | 56.78 | 64.60 | 39.63 |
| CLO | 44.28 | 36.50 | 45.78 | 48.96 | 44.82 | 57.89 | 31.71 |
| + EN-steering * | 42.98 | 35.69 | 44.97 | 47.22 | 43.11 | 55.45 | 31.44 |
| + LOC-steering * | 45.41 | 37.07 | 47.13 | 51.22 | 45.88 | 58.90 | 32.27 |
| + LOC-steering ** | 51.86 | 44.95 | 51.75 | 59.81 | 55.40 | 62.37 | 36.89 |
| + SUR-steering | 50.94 | 44.19 | 50.68 | 58.51 | 54.48 | 61.51 | 36.29 |
| MIDALIGN | 44.95 | 39.67 | 45.65 | 49.44 | 42.46 | 58.07 | 34.39 |
| + EN-steering * | 44.15 | 39.54 | 44.66 | 48.87 | 42.08 | 56.26 | 33.52 |
| + LOC-steering * | 46.52 | 40.38 | 47.94 | 51.57 | 44.16 | 59.59 | 35.47 |
| + LOC-steering ** | 52.07 | 47.32 | 51.45 | 59.40 | 53.13 | 62.75 | 38.40 |
| + SUR-steering | 51.57 | 46.60 | 51.15 | 58.47 | 53.39 | 62.39 | 37.43 |

