# OpenReview forum: "Rethinking Cross-lingual Alignment: Balancing Transfer and Cultural Erasure in Multilingual LLMs"
_ICLR.cc/2026/Conference — Submitted to ICLR 2026_

### Official Review · Reviewer_73cB · 2025-10-31

**Soundness:** 2
**Presentation:** 3
**Contribution:** 4
**Rating:** 8
**Confidence:** 4

**Summary:**

Under the assumption that language represents culture, this paper investigates the dual effects of cross-lingual alignment on universal and culture-specific knowledge in LLMs. This paper proposes a holistic evaluation framework that comprehensively assesses how LLMs’ mastery of shared and culture-specific knowledge changes across languages before and after cross-lingual alignment. The results reveal a clear trade-off between the two. Furthermore, the paper examines the representation spaces and introduces a representation engineering (RE) approach, in which language alignment and cultural differentiation REs are applied at different layers. Taken together, these two investigations can provide insights for future research.

Two main limitations remain. First, the discovered trade-off is validated on only one model and a pair of datasets (one for universal knowledge and the other for culture-specific knowledge evaluation). Second, the assumption that language represents culture is overly strong, which further weakens the validity of the representation explorations.

**Strengths:**

- The topic (intersection of multilingual and culture) is practical; and the research question (relationship between cross-lingual alignment and cultural alignment) is intuitive and important.

- The methodology is straightforward and easy to follow: applying cross-lingual alignment (CLA) techniques to LLMs and evaluating changes in the models’ mastery of shared and culture-specific knowledge in multilingual contexts.

- The experimental results are intuitive: although CLA improves the cross-lingual transfer of shared knowledge, it leads to erasure of cultural-specific knowledge, revealing a clear trade-off between cross-lingual transfer and cultural preservation.

- The representation analysis in Section 5 is interesting — it identifies distinct layer distributions for shared versus cultural knowledge. The proposed representation engineering (RE) method based on this observation offers useful insights for mitigating this trade-off in future work.

**Weaknesses:**

- Despite testing different CLA methods and languages, the experiments are still limited: results are based only on a single model (Gemma3 12B), with shared knowledge evaluated solely on the GMMLU dataset and cultural-specific knowledge on BLEND.

- The paper adopts a strong language-as-culture-proxy assumption and removes explicit localization context from BLEND (Section 3). This assumption is debatable, yet its rationale and limitations are not explicitly discussed.

- The motivation behind the representation analysis in Sections 4.4 and 5 is insufficient. To directly test the question, “If CLA suppresses a model’s ability to use language as a cultural cue, is that knowledge permanently erased or merely inaccessible?”, the most straightforward approach would be to reintroduce explicit cultural localization cues in the prompts (as done by Veselovsky et al., 2025). Without such comparisons, the effectiveness of the proposed RE method is also not fully validated.

**Questions:**

- The GMMLU dataset does not appear to be a parallel multilingual dataset and includes cultural nuances — does this affect the validity of the results?
- It would be better if the representation similarity in Section 4.4 is quantitatively measured.
- How to further explain the phenomenon in Section 4.4? Is it just because BLEND datasets are more surface-level different across languages, or does the LLMs genuinely encode shared and cultural knowledge in different layers?
- Could more details be provided on EN-steering and LOC-steering? For example, from which datasets are they derived? (From Figure 4b, it seems both datasets were used for vector extraction)

---

> ### Author Response · Authors · 2025-11-23
> **Rebuttal Response to Reviewer 73cB (1/2)**
>
> We thank the reviewer for the constructive and supportive feedback. We especially appreciate the acknowledgment that the contribution is excellent, and the research question is intuitive and important. Our detailed responses follow. We would gladly engage in further discussion.​​
>
> > W1. Despite testing different CLA methods and languages, the experiments are still limited: results are based only on a single model (Gemma3 12B), with shared knowledge evaluated solely on the GMMLU dataset and cultural-specific knowledge on BLEND.
>
> We address your question about the generalization on other models and datasets. For completeness, we summarize the key findings here and provide full details in our General Response section. To directly evaluate generalizability, we conducted additional experiments on a new model (Qwen2.5 14B) and a new culturally specific dataset (CulturalBench). These results show that (1) the trade-off between knowledge transfer and cultural localization persists across architectures and datasets, and (2) SUR-steering consistently achieves a better balance between the two dimensions in all settings we tested.
>
> ***
>
> > W2. The paper adopts a strong language-as-culture-proxy assumption and removes explicit localization context from BLEND (Section 3). This assumption is debatable, yet its rationale and limitations are not explicitly discussed.
>
> As you pointed out, we have that strong language-as-cultural-proxy assumption, and we should have discussed more explicitly.
>
> What is often discussed in cross-lingual alignment and knowledge transfer focuses on surface linguistic forms (lexical or syntactic alignment), while discussions of cultural knowledge tend to be more fine-grained and region-specific, even within the same language. Because our work aims to connect these two perspectives in one framework, we begin with a simplified assumption: that surface linguistic cues can serve as implicit cultural cues. This is an abstraction, and we acknowledge it as a limitation, but it allows us to take a first, tractable step toward studying how alignment affects culturally grounded reasoning across languages.
>
> We will add this discussion of rationale and limitations in the future draft, and we agree that future work should incorporate richer signals of cultural grounding beyond linguistic form alone.
>
>
> ***
>
> > W3. The motivation behind the representation analysis in Sections 4.4 and 5 is insufficient. To directly test the question, “If CLA suppresses a model’s ability to use language as a cultural cue, is that knowledge permanently erased or merely inaccessible?”, the most straightforward approach would be to reintroduce explicit cultural localization cues in the prompts (as done by Veselovsky et al., 2025). Without such comparisons, the effectiveness of the proposed RE method is also not fully validated.
>
> Thank you for pointing that out. We experimented by reintroducing the explicit localization context into the prompt on MIST and a more strongly aligned model, MidAlign. For both models, adding explicit context substantially improves BLEND accuracy, but it also causes severe degradation on GlobalMMLU (Table D). Notably, the BLEND gains from this explicit prompting are not larger for the more aligned MidAlign model compared to MIST.
> These results confirm that cultural knowledge is not permanently erased; rather, CLA methods suppress the model’s ability to use language as a cultural cue. This suppressed information can be reactivated effectively through SUR-steering, which restores access to localized knowledge while still benefiting from improved transfer.
>
> **Table D: Explicit context prompting experiment on MIST and MidAlign.**
> | Gemma3 | GMMLU(%) |  | BLEND(%) |  | Gemma3 | GMMLU(%) |  | BLEND(%) |  |
> |---|:---:|---|:---:|---|---|:---:|---|:---:|---|
> | MIST | 59.74 |  | 46.90 |  | MidAlign | 61.18 |  | 44.95 |  |
> | MIST with Explicit Context Prompt  | 58.11 | -1.63 | 79.54 | 32.64 | MidAlign with Explicit Context Prompt  | 59.64 | -1.54 | 80.28 | 35.33 |
> | EN-Steering | 59.90 | 0.16 | 46.45 | -0.45 | EN-Steering | 61.44 | 0.26 | 44.15 | -0.80 |
> | LOC-Steering | 59.83 | 0.09 | 54.53 | 7.63 | LOC-Steering | 61.04 | -0.14 | 52.07 | 7.12 |
> | SUR-Steering | 60.07 | 0.33 | 54.37 | 7.47 | SUR-Steering | 61.20 | 0.02 | 51.57 | 6.62 |

---

> ### Author Response · Authors · 2025-11-23
> **Rebuttal Response to Reviewer 73cB (2/2)**
>
> > Q1. The GMMLU dataset does not appear to be a parallel multilingual dataset and includes cultural nuances — does this affect the validity of the results?
>
> GlobalMMLU is a multi-way translation of MMLU, featuring extended annotations for cultural specificity. All 14k instances in the test set are parallel across languages. While it includes culturally sensitive (CS) instances, their proportion is relatively small (792) compared to the full test set. We evaluated this by dividing the GlobalMMLU test set into CS (792) and non-CS (13.2k) subsets. Performance on the non-CS subset remained largely unchanged, whereas the CS subset showed different trends for LOC and SUR. LOC steering tended to yield higher Gmmlu-CS set performance, and SUR LOC results became much closer. This indicates that localization helps in retrieving culturally specific knowledge.
>
> We include this additional plot in Figure 17 of the updated pdf version.
>
> ***
>
> > Q2. It would be better if the representation similarity in Section 4.4 is quantitatively measured.
>
> To quantitatively measure the representational similarity in Section 4.4, we use a simple k-NN cosine-similarity–based score. For each activation, we compute its cosine similarity to all other activations (both from the same language and from different languages), take the top-k nearest neighbours (k = 10), and measure what fraction of those neighbours come from other languages. This gives a direct estimate of how intermingled language-specific activation clusters are. Higher values indicate that activations from different languages are strongly mixed, suggesting cross-lingual alignment is merging their representational manifolds, whereas lower values indicate that languages remain more geometrically separable in hidden space.
>
> **Table E: Quantifying cross-language representation similarity per layer.**
> | GlobalMMLU | Unaligned | MIST | CLO | MidAlign | BLEND | Unaligned | MIST | CLO | MidAlign |
> |---|:---:|:---:|:---:|:---:|:---:|:---:|:---:|:---:|:---:|
> | Layer 20 | 0.619 | 0.675 | 0.768 | 0.758 | Layer 20 | 0.107 | 0.125 | 0.346 | 0.218 |
> | Layer 28 | 0.699 | 0.756 | 0.802 | 0.824 | Layer 28 | 0.299 | 0.344 | 0.308 | 0.450 |
> | Layer 46 | 0.255 | 0.297 | 0.045 | 0.539 | Layer 46 | 0.122 | 0.124 | 0.004 | 0.221 |
>
> The result in Table E aligns with the observation in Section 4.4 that, as cross-lingual alignment becomes stronger, language representations converge more tightly, and that GlobalMMLU begins to merge in the middle layers, whereas BLEND remains largely separable until much deeper layers, even after CLA.
>
>
> ***
>
> > Q3. How to further explain the phenomenon in Section 4.4? Is it just because BLEND datasets are more surface-level different across languages, or does the LLMs genuinely encode shared and cultural knowledge in different layers?
>
> We hypothesize it is the latter. It is difficult to draw strong conclusions from PCA alone, but across both Gemma3 (Figure 4b, Figure 9) and Qwen2.5 (Figure 15) in layer-wise performance analysis on BLEND, we observe clear layer-specific “peaks” where localization steering is most effective. This may suggest that cultural or localization-related information is encoded more strongly in specific layers rather than being simply language differences in BLEND.
>
>
> ***
>
> > Q4. Could more details be provided on EN-steering and LOC-steering? For example, from which datasets are they derived? (From Figure 4b, it seems both datasets were used for vector extraction)
>
> For En-steering, we first use layer-wise Principal Component Analysis (Wold et al., 1987,PCA) analysis and identify layers 16-32 to exhibit the highest overlap of hidden activations across languages which is necessary for steering to be effective (Section 4.2 ,L247–L251). We then extract the steering vectors from the activations of the GlobalMMLU dev1 set. We apply EN-steering at layer 20 based on the accuracy on the GlobalMMLU dev2 set (Figure 4b).
> For LOC-steering, we extract localization vectors from activations of the BLEND dev1 set. A layer-wise analysis on BLEND dev2 shows that LOC-steering peaks around layer 28, so we apply localization steering at that layer.
> Thus, each steering vector is extracted from the dataset that best corresponds to its purpose: GlobalMMLU for EN-steering and BLEND for LOC-steering.

---

### Official Review · Reviewer_NjxX · 2025-10-31

**Soundness:** 3
**Presentation:** 3
**Contribution:** 2
**Rating:** 4
**Confidence:** 4

**Summary:**

The study aims to disentangle knowledge transfer and cultural knowledge erasure. They propose an inference-time method to steer activations for knowledge and cultural information at different transformer layers. The study finds that middle layers work best for knowledge transfer, while cultural information is encoded in deeper layers.
Knowledge is categorized into universal knowledge (language agnostic knowledge) with the objective of best possible transfer and culturally-adaptive knowledge based on universal concepts but different instantiation through cultural norms, regulation etc., with the objective of best possible localization. The proposed method is evaluated on global mmlu and BLEND. The paper proposes SUR-steering method that applies localized activation steering inspired by related steering and representation alignment methods.

**Strengths:**

- The study is well motivated
-  The evaluation results of the proposed steering method show that localization can be achieved without negatively affecting global mmlu performance
- The paper makes a good contribution by providing a unified view on knowledge transfer and cultural knowledge erasure

**Weaknesses:**

- It's unclear how multilingual instruction-tuning, midalign, and steering are encouraging localization, as they map target languages to English representation space
- The proposed SUR-steering method lacks innovation as it applies existing steering to existing cross-lingual alignment methods. While the insights are interesting, performance differences are low for the best performing methods (MidAlign, CLO)
- The underlying technical mechanisms negatively affecting cultural knowledge are not discussed in detail

**Questions:**

- How could the benefits of SUR-steering be applied when training multilingual models?
- Knowledge localization extends beyond languages, i.e. user can ask for emergency number in a different country. How to adapt the method to localize knowledge beyond language?
  - Different languages are an arbitrary boundary for localized knowledge
- typo line 97

---

> ### Author Response · Authors · 2025-11-23
> **Rebuttal Response to Reviewer NjxX (1/2)**
>
> Thank you for your remarks that our work is well-motivated and provides a solid contribution. We appreciate your insightful and careful evaluation. Below, we address each of your comments in turn. We would gladly engage in further discussion.​​
>
> > W1. It's unclear how multilingual instruction-tuning, midalign, and steering are encouraging localization, as they map target languages to English representation space.
>
>
> To clarify, multilingual instruction-tuning, midalign, and en-steering are encouraging “knowledge transfer”, by mapping target language to English representation space. If you meant to ask why these methods “discourage” localization, our view is as follows: when the model’s internal representations are aligned toward English, it indeed becomes better at transferring universal knowledge, but this alignment also pushes the model toward English-centric interpretations. As a result, the model loses some of the culturally grounded distinctions and language-specific cues that were originally present, leading to functional degradation on questions whose correct answers depend on localized cultural knowledge. In this sense, representational alignment toward English improves transfer but simultaneously erases part of the localized knowledge.
>
> If this is not what you intended to ask, we would greatly appreciate clarification.
>
> ***
>
> > W2. The proposed SUR-steering method lacks innovation as it applies existing steering to existing cross-lingual alignment methods. While the insights are interesting, performance differences are low for the best performing methods (MidAlign, CLO)
>
> We acknowledge that the technical method itself is not entirely new. The novelty of our approach is in how we use different types of steering vectors in a complementary and targeted way to address the transfer–localization trade-off.
> While prior work applies a single type of steering vector (English or Localization) to improve a one-sided target, our insight is that different steering vectors affect transfer and localization in distinct ways, and therefore combining them at different layers can achieve a better balance than either alone. SUR-Steering operationalizes this idea by applying EN-steering at transfer-sensitive layers and LOC-steering at localization-sensitive layers.
>
> As pointed out, models with stronger alignment, such as MidAlign and CLO, are less responsive to steering than the unaligned or MIST models. We hypothesize that the fundamental trade-off persists: while some cultural knowledge can be partially recovered through steering, certain cultural nuances may be irreversibly lost during the alignment training process. This is also observed in our analysis of English-biased responses: although steering mitigates the bias to some extent, the post-trained aligned models never fully return to the  unaligned baseline, suggesting inherent limits to what steering can recover (Figure 5).
>
> ***
>
> > W3. The underlying technical mechanisms negatively affecting cultural knowledge are not discussed in detail.
>
> To restate our understanding of the question: why cross-lingual alignment techniques negatively affect cultural knowledge, and what underlying mechanisms lead to this loss.
> As discussed in Section 4.4, we analyze how cross-lingual alignment (CLA) reshapes the model’s internal representation space (Figure 3). Our analysis shows that CLA induces stronger representational convergence, particularly in the deeper layers where localization steering is most sensitive. This convergence reduces the distinctiveness of culturally grounded representations, helping to explain why cultural knowledge is negatively affected by alignment.

---

> ### Author Response · Authors · 2025-11-23
> **Rebuttal Response to Reviewer NjxX (2/2)**
>
> > Q1. How could the benefits of SUR-steering be applied when training multilingual models?
>
> MIST, CLO, and MidAlign are post-training cross-lingual alignment CLA methods, whereas steering operates at inference time. This means that once training is complete, we can apply the same steering procedure—extracting steering vectors and injecting them during inference—to any of these CLA-trained models.
> As results, applying SUR-steering to models already trained with MidAlign and CLO yields further improvements in both transfer and localization. The gain by SUR-steering with those “already aligned” models is smaller, however, suggesting a saturation effect in models that already possess high transfer capabilities.
>
>
> ***
>
> > Q2. Knowledge localization extends beyond languages, i.e. user can ask for emergency number in a different country. How to adapt the method to localize knowledge beyond language?
>
> Our work adopts a language-as-cultural-proxy assumption for simplicity, but we agree that cultural knowledge extends beyond language. There are, however, possible extensions of our method that would allow localization beyond language identity.
>
> Unlike EN-steering, LOC-steering vectors can be computed from different cultural contexts within the same language (e.g., “in Spain” vs. “in Mexico” for Spanish). In principle, this would allow the model to steer toward a specific regional or cultural context independently of the input language. The most accurate approach would be to select the appropriate LOC-steering vector conditionally based on additional metadata (such as country or region), when such metadata is available. However, in many real-world settings, explicit localization metadata may not be provided or could be incorrect. So, linguistic surface form could be implicit cues as we assume in our work. Another approach could be to aggregate LOC-steering vectors across multiple regions that share the same language, for example, by averaging region-specific vectors, to create a semi-generalized localization vector per language that captures common culturally grounded knowledge beyond language alone.
>
>
> ***
>
> > Q3. typo line 97
>
> Thank you for the comment! We updated the pdf, fixing that.

---

### Official Review · Reviewer_uUYH · 2025-11-01

**Soundness:** 3
**Presentation:** 4
**Contribution:** 4
**Rating:** 6
**Confidence:** 4

**Summary:**

The paper investigates how cross-lingual alignment in multilingual models improves factual transfer but causes cultural erasure. Using the proposed transfer-localization plane, the authors reveal this trade-off across six languages. Moreover, they introduce Surgical Steering, an inference-time method that targets different model layers to better balance factual transfer and cultural localization.

**Strengths:**

1. This paper addresses a compelling topic—the trade-off between knowledge transfer and cultural localization—pioneering research in this area. By exploring this trade-off, multilingual alignment can achieve better semantic consistency across languages while preserving culturally relevant differences.
2. The paper offers deep insights into cross-lingual alignment through the transfer-localization plane, demonstrating that current multilingual methods enhance transfer performance at the cost of cultural localization, highlighting opportunities for improvement.
3. The authors propose Surgical Steering, an inference-time method that adjusts model representations to achieve a better balance between knowledge transfer and cultural localization.

**Weaknesses:**

1. The insights and contributions of this paper are impressive. However, the evaluation is limited to a single model and one culturally specific dataset. It would be important to demonstrate the method’s generalizability across other models and datasets.
2. The proposed Surgical Steering appears to be an adaptation of existing English Steering methods, which raises questions about presenting it as a wholly novel approach for cross-lingual alignment. Nevertheless, the authors provide a valuable insight into the trade-off between transfer and cultural localization in multilingual models.

**Questions:**

1. Do you have results on other models and cultural-specific datasets?
2. How do you calculate the LOC-steering vector? I didn't find a clear explanation for it in the paper.

---

> ### Author Response · Authors · 2025-11-23
> **Rebuttal Response to Reviewer uUYH**
>
> We truly appreciate your recognition of the work as impressive and pioneering research in this area. Thank you for the thoughtful and constructive review.  Below, we address each of your comments in detail.​​
>
> > W1&Q1. The insights and contributions of this paper are impressive. However, the evaluation is limited to a single model and one culturally specific dataset. It would be important to demonstrate the method’s generalizability across other models and datasets.
>
> We address your question about the generalization on other models and datasets. For completeness, we summarize the key findings here and provide full details in our General Response section. To directly evaluate generalizability, we conducted additional experiments on a new model and a new culturally specific dataset. These results show that (1) the trade-off between knowledge transfer and cultural localization persists across architectures and datasets, and (2) SUR-steering consistently achieves a better balance between the two dimensions in all settings we tested.
>
>
> ***
>
> > W2. The proposed Surgical Steering appears to be an adaptation of existing English Steering methods, which raises questions about presenting it as a wholly novel approach for cross-lingual alignment. Nevertheless, the authors provide a valuable insight into the trade-off between transfer and cultural localization in multilingual models.
>
> You are right that SUR-Steering builds on existing Steering approaches, and we acknowledge that the mechanism itself is not entirely new. What we believe is novel is how we use different types of steering vectors in a complementary and targeted way to address the transfer–localization trade-off.
>
> While prior work applies a single type of steering vector (English or Localization) to improve a one-sided target, our insight is that different steering vectors affect transfer and localization in distinct ways, and therefore combining them at different layers can achieve a better balance than either alone. SUR-Steering operationalizes this idea by applying EN-steering at transfer-sensitive layers and LOC-steering at localization-sensitive layers.
>
> ***
>
>
> > Q2. How do you calculate the LOC-steering vector?
>
> Thank you for the clarification question. The LOC-steering vectors are extracted from paired inputs with and without cultural context ($x_{\text{con}}$ and $x_{\text{decon}}$). These pairs come from 100 samples of the BLEND dev1 set (Section 5.1, L342–L350). $x_{\text{con}}$ is the original BLEND sentence that contains an explicit localization phrase (e.g., “in Greece”). $x_{\text{decon}}$ is the decontextualized version where that explicit localization phrase is removed (Section 3, L160–L161; Appendix A.2).
>
> For each pair, we collect the hidden activations $h(x_{\text{con}})$ and $h(x_{\text{decon}})$ at layer $l$. The LOC-steering vector is then obtained by computing the average activation difference across all such pairs, following the same procedure used for English-Steering (Section 4.1, L193–L201).

---

### Official Review · Reviewer_cuHY · 2025-11-07

**Soundness:** 1
**Presentation:** 3
**Contribution:** 1
**Rating:** 2
**Confidence:** 4

**Summary:**

This paper is about the dark side of cross-lingual alignment or transfer. The authors conducted empirical studies for four common cross-lingual alignment frameworks (CLA)  to observe culture erasure after training, and they suggest considering both transfer/alignment and localization in CLA. Then, the authors propose a representation patching method that surgically steers shallow layers to “English” and middle-to-deep layers to the “target” language.

**Strengths:**

1.	The presentation is clear.

2.	The authors provide some valuable, negative results for CLA.

**Weaknesses:**

1.	While the negative results are valuable, the overall contribution of the paper is not sufficient. The authors re-run existing frameworks on existing datasets, and the findings are not new. For example, the cultural bias is discussed in GLOBAL_MMLU, which the authors consider and use.

2.	The work is not complete. The method, which relies on surgery of the model, is not easy to reproduce on other models. The authors only conduct experiments on one model, making the generality unclear. Is there a principled way or an automated way to configure the method? Do you have any ablation studies for this?

**Questions:**

Please refer to Weaknesses

---

> ### Author Response · Authors · 2025-11-23
> **Rebuttal Response to Reviewer cuHY**
>
> We appreciate your insightful feedback and thank you for noting that our paper provides valuable results. Below, we provide detailed responses to each of your points and would be happy to offer further clarification if needed.
>
> > W1. While the negative results are valuable, … authors re-run existing frameworks on existing datasets, and the findings are not new. For example, the cultural bias is discussed in GLOBAL_MMLU, which the authors consider and use.
>
> To date, research on cross-lingual transfer and culturally-situated models has largely proceeded in isolation. Our contribution is on unifying these two strands with a framework designed to uncover the hidden costs of alignment and to develop interventions that balance shared knowledge with cultural specificity.
>
> Specifically, we do not simply "reveal" the presence of cultural bias (as discussed in GlobalMMLU and many other works); we also provide evidence of its origin by showing that the pursuit for cross-lingual alignment in linked to erasure of localized knowledge (Section 4.3 and Figure 5b). Moreover, our findings demonstrate that the proposed steering strategies can achieve a better balance between factual transfer and cultural localization.
>
> ***
>
> > W2. The method, which relies on surgery of the model, is not easy to reproduce on other models. The authors only conduct experiments on one model, making the generality unclear. Is there a principled way or an automated way to configure the method?
>
> We address the reviewer’s concern about the generalization of our method to other models. For clarity, we provide a brief summary here and refer you to our General Response for full details. We conducted additional experiments on a new model to directly assess generalizability. Across these experiments, SUR-steering consistently offers a better balance between knowledge transfer and cultural localization, indicating that the approach extends beyond the specific model used in the main paper. We also provide clear, reproducible steps for configuring the method in our general response.
>
> >> W2-2. Do you have any ablation studies for this?
>
> Yes. An ablation study for surgical steering corresponds to evaluating each component in isolation: EN-steering alone, LOC-steering alone, and the unmodified base model. These comparisons already appear in our results (each steering is reported individually). If this does not match what you intended, we would appreciate clarification so we can address the ablation analysis more precisely in the revision.

---

### Author Response · Authors · 2025-11-23
**General Response (1/2)**

We sincerely appreciate the reviewers’ constructive comments and feedback. Below, we address a common concern about the generalization of the findings raised by the reviewers.

[TL;DR]
We conducted additional experiments to address concerns regarding generalizability, evaluating both the trade-off and the proposed mitigation strategy on a new model and a new culturally specific dataset. We include the corresponding tables here in response, and have also updated the paper with the new plots in Appendix Figures 15 and 16.

We show that:
 (1) the trade-off between knowledge transfer and cultural localization generalizes across architectures and cultural datasets, and
(2) our proposed surgical steering (SUR-steering) consistently achieves a better balance between the two competing dimensions across models and datasets.


> [Reviewer cuHY] The method, which relies on surgery of the model, is not easy to reproduce on other models. The authors only conduct experiments on one model, making the generality unclear.

> [Reviewer uUYH] The insights and contributions of this paper are impressive. However, the evaluation is limited to a single model and one culturally specific dataset. It would be important to demonstrate the method’s generalizability across other models and datasets.

> [Reviewer 73cB] Despite testing different cross-lingual alignment (CLA) methods and languages, the experiments are still limited: results are based only on a single model (Gemma3 12B), with shared knowledge evaluated solely on the GMMLU dataset and cultural-specific knowledge on BLEND.
***
**Generalizability of trade-off and SUR-Steering across Models**
We conduct further experiments to show that the trade-off between knowledge transfer and cultural localization and our proposed mitigation method generalizes across models.

We conduct our experiments on Qwen2.5-14B using the same setup as in the Gemma experiments. First, following Section 4.2 (L247–L251), we use layer-wise Principal Component Analysis (Wold et al., 1987,PCA) and identify layers 20-36 to exhibit the highest overlap of hidden activations across languages which is necessary for steering to be effective. We apply EN-steering at layer 24 based on the accuracy on the GlobalMMLU development set (dev2). Then, similarly in Section 5.2 layer-wise analysis on dev2 of BLEND, we observe that LOC-steering peaks around layer 32, and therefore apply LOC-steering at that layer. We provide additional plots of this layer-wise analysis in Appendix Figure 15 for both the development and the test set. Finally, we apply surgical steering, combining EN-steering at the earlier layer and LOC-steering at the deeper layer.

Additionally, we newly trained multilingual instruction tuning (MIST) with Qwen model as one of the cross-lingual alignment post-training methods. Unlike the full-parameter training used in the main paper, this version was trained using LoRA (Hu et al., 2022). We used the same dataset for one epoch as in the main experiments. We will include full training details in the revised version of the paper. Results are given below:

**Table A: Generalizability of  trade-off and SUR-Steering on Qwen2.5 model.**
| Qwen | GMMLU (%) |  | BLEND (%) |  |  | Gemma | GMMLU (%) |  | BLEND (%) |  |
|:---:|:---:|---|:---:|---|---|:---:|:---:|---|:---:|---|
| Unaligned | 62.74 |  | 51.09 |  |  | Unaligned | 58.86 |  | 47.64 |  |
| +EN-Steering | 63.05 | +0.31 | 49.85 | -1.24 |  | +EN-Steering | 59.14 | +0.28 | 47.04 | -0.60 |
| +LOC-Steering | 62.11 | -0.63 | 54.91 | +3.82 |  | +LOC-Steering | 59.16 | +0.30 | 55.27 | +7.63 |
| +SUR-Steering | 63.01 | +0.27 | 51.4 | +0.31 |  | +SUR-Steering | 59.39 | +0.53 | 55.44 | +7.80 |
| MIST | 63.24 | +0.50 | 49.52 | -1.57 |  | MIST | 59.74 | +0.88 | 46.9 | -0.74 |
| +EN-Steering | 63.43 | +0.69 | 48.72 | -2.37 |  | +EN-Steering | 59.9 | +1.04 | 46.45 | -1.19 |
| +LOC-Steering | 62.51 | -0.23 | 53.19 | +2.10 |  | +LOC-Steering | 59.83 | +0.97 | 54.53 | +6.89 |
| +SUR-Steering | 63.45 | +0.71 | 50.27 | -0.82 |  | +SUR-Steering | 60.07 | +1.21 | 54.37 | +6.73 |

First, applying cross-linual alignment methods (here, EN-steering and MIST) increases knowledge-transfer performance on GlobalMMLU while simultaneously erasing cultural localization knowledge, confirming that the trade-off continues to hold on a different model.
Unlike the Gemma model, Qwen’s Localization-Steering (LOC-steering) exhibits a substantial drop in GlobalMMLU accuracy, which further underscores the trade-offs.
 On top of that, surgical steering provides consistently better trade-offs: it largely preserves EN-steering’s improvements on the GlobalMMLU side, while not only recovering the lost localized knowledge but also further improving from its baseline. This results in a more favorable balance between knowledge transfer and cultural localization and demonstrates that surgical steering can be applied beyond a single model.
The trade-off plot is available in Appendix Figure16.

---

> ### Author Response · Authors · 2025-11-23
> **General Response (2/2)**
>
> **Generalizability across Datasets**
> We show the transfer-localization trade-offs generalized on other culturally specific datasets, and our proposed method can be generally effective to those datasets as well.
>
> To that end, we conduct additional experiments on CulturalBench (Chiu et al., ACL 2025). Similarly to BLEND, we tailor the dataset to our needs, by automatically generating a decontextualized version by removing explicit localization context from its questions (detailed in Section3 and Appendix A.2). Unlike BLEND, which provides questions directly in the source language, CulturalBench is available only in English, so we translate every question-options set into the target language using the same translation prompt used in BLEND option translation. Note that there are only four regions and languages overlapping with the ones we focused on in our paper, and the extracted sample size was much smaller than BLEND (ES: 40, ID: 26, KO: 41, ZH: 59). Since the overall sample size is small, we use the same subset for extracting the steering vectors.
>
> ***
> **Transfer-Erasure Trade-offs Generalize to Different Cultural Datasets**
> Table B demonstrates that the trade-off is not specific to BLEND: the same pattern persists when evaluated on CulturalBench. Despite the smaller sample size, the same trend emerges: methods that improve cross-lingual knowledge transfer tend to erase cultural localization. We report the average accuracy of four language set. The base model is Gemma3, the same model reported with BLEND.
>
> **Table B: Trade-offs between GlobalMMLU and CulturalBench**
> | Gemma3 | GMMLU (%) |       | CULTURALBENCH (%) |       |
> |:---------:|:---------:|:-----:|:-----------------:|:-----:|
> | Unaligned |   59.63   |       |       61.29       |       |
> |    MIST   |   60.55   | +0.92 |       60.32       | -0.97 |
> |    CLO    |    61.6   | +1.97 |       56.28       | -5.01 |
> |  MidAlign |   61.86   | +2.23 |       56.61       | -4.68 |
>
> ***
> **SUR-Steering Trade-off Improvements Generalize to Different Cultural Datasets**
> Table C evaluates whether SUR-Steering generalizes beyond BLEND by applying it to CulturalBench. While LOC-steering attempts to recover the localized knowledge that was there in the unaligned model but erased by the cross-lingual alignment methods (CLO and MidAlign), this recovery often comes at the cost of reduced knowledge-transfer performance. Conversely, EN-steering tends to improve transfer but typically comes with the slight erasure of localized knowledge. SUR-steering offers a better balance: it preserves the transfer gains of EN-steering while still recovering much of the localized knowledge lost under CLO and MidAlign. Notably, for CLO, SUR-steering even surpasses LOC-steering on CulturalBench, demonstrating its stronger ability to restore localized knowledge without largely sacrificing transfer.
>
> **Table C: Generalizability of SUR-Steering on CulturalBench (CULBENCH)**
> | Gemma3 | GMMLU (%) |  | CULBENCH (%) |  | Gemma3 | GMMLU (%) |  | CULBENCH (%) |  |
> |:---:|:---:|:---|:---:|:---|:---:|:---:|:---|:---:|:---|
> | MidAlign | 61.86 |  | 56.61 |  | CLO | 61.60 |  | 56.28 |  |
> | +EN-Steering | 62.05 | +0.19 | 56.19 | -0.42 | +EN-Steering | 61.86 | 0.26 | 56.29 | +0.01 |
> | +LOC-Steering | 61.65 | -0.21 | 58.62 | +2.01 | +LOC-Steering | 61.20 | -0.4 | 57.48 | +1.20 |
> | +SUR-Steering | 61.96 | +0.10 | 58.51 | +1.90 | +SUR-Steering | 61.79 | 0.19 | 58.93 | +2.65 |
>
> ***
>
> [Reference]
> Qwen2.5: A Party of Foundation Models. (Qwen team, 2024)
> CulturalBench: A Robust, Diverse and Challenging Benchmark for Measuring LMs’ Cultural Knowledge Through Human-AI Red-Teaming (Chiu et al., ACL 2025)
> LoRA: Low-Rank Adaptation of Large Language Models (Hu et al., ICLR 2022)

---

### Author Response · Authors · 2025-12-03
**Rebuttal Summary**

Thank you for taking the time to review our work, especially under these special review conditions.

As the discussion period comes to an end, we would like to thank the reviewers for their constructive and encouraging feedback. We especially appreciate the recognition that our work represents *pioneering research in this area* and that *the insights and contributions of this paper are impressive* [Reviewer uUYH]. We are also grateful for comments noting that the contribution is *excellent* and the research question *intuitive and important* [Reviewer 73cB], as well as the acknowledgment that the study is *well motivated* and makes a *good contribution* [Reviewer NjxX] with *valuable results* [Reviewer cuHY].


Although the shortened rebuttal period limited reviewer engagement, we believe our responses have thoroughly addressed the raised concerns and questions. Below, we summarize the key contributions, main concerns, and how we addressed them.


***
## Main Contribution of Our Work

1. **Novel framing of a key trade-off of transfer and cultural erasure.** The paper reframes cross-lingual alignment (CLA) by showing that aligning multilingual representations for better knowledge transfer comes at the cost of “cultural erasure.” This trade-off perspective (transfer vs. cultural localization) is rarely explored, where each component has largely proceeded in isolation.
2. **Comprehensive evaluation across methods and languages confirms the trade-off.** Using the “transfer-localization plane,” we evaluate multiple CLA approaches across several languages, empirically showing that the pursuit for cross-lingual alignment is linked to the erasure of localized knowledge.
3. **Proposal of a practical mitigation method: “Surgical Steering.”** By disentangling transfer and localization based on layer-wise analysis, we propose an inference-time method by steering different kinds of activations at different layers, achieving a better balance between the two competing dimensions.


***
## Addressing Reviewer Concerns


1. **Strong empirical evidence of generalizability.**
    - As detailed in the General Comment, our findings of the trade-off and the proposed approach are validated, not just on a single model or dataset, but across different models and culturally specific datasets.
2. **Clarification of Contributions.**
    - Our work makes a distinct contribution by unifying two previously isolated research areas: cross-lingual alignment and culturally-situated models. We move beyond simply noting the presence of cultural bias (as discussed in previous works) to empirically establishing a causal link between the pursuit of cross-lingual alignment and the erasure of localized knowledge. This framing of the transfer-localization trade-off is novel.
    - Our contribution on the Surgical Steering approach (SUR-Steering) lies in showing that different types of steering vectors influence transfer and localization in targeted ways, and that combining them at different layers yields a better balance than using either alone. SUR-Steering operationalizes this insight by applying English steering at transfer-sensitive layers and Localization steering at localization-sensitive layers, even though the underlying steering mechanism itself is not new.
3. **Enhanced Clarity and Detail.**
    - GlobalMMLU is fully parallel across 14k items, and culturally sensitive questions make up only a small portion. The presence of some culturally nuanced items in GlobalMMLU does not invalidate our existing findings based on our separation analysis.
    - Details of the experiments, including clear steps for reproducing SUR-steering, are provided in the individual reviewer responses and in the general response.

---

### Meta-Review · Area_Chair_Wq2B · 2026-01-13

**Summary:**

The reviewers generally agree that the paper studies an important and timely question: the trade-off between cross-lingual knowledge transfer and cultural localization in multilingual LLMs. Several reviewers acknowledge that the problem formulation is intuitive and practically relevant, and that the paper is clearly written with a coherent experimental pipeline (e.g., Reviewer uUYH: “addresses a compelling topic”; Reviewer 73cB: “the research question is intuitive and important”; Reviewer cuHY: “the presentation is clear”).

However, the main concerns that informed my decision are threefold.
First, there is substantial disagreement regarding the novelty and contribution of the work. Multiple reviewers argue that the paper largely re-runs existing cross-lingual alignment and steering techniques on existing benchmarks, and that the proposed Surgical Steering method is an adaptation or combination of known steering approaches rather than a fundamentally new method (Reviewer cuHY: “overall contribution is not sufficient”; Reviewer NjxX: “applies existing steering to existing cross-lingual alignment methods”; Reviewer uUYH: “appears to be an adaptation of existing English Steering methods”).

Second, although the authors add additional experiments in the rebuttal, reviewers raise concerns about the strength and generality of the central assumptions. In particular, the work relies heavily on a language-as-culture-proxy assumption, which several reviewers find overly strong or insufficiently justified, limiting the scope of the conclusions (Reviewer 73cB: “the assumption that language represents culture is overly strong”; Reviewer NjxX: “different languages are an arbitrary boundary for localized knowledge”).

Third, while the rebuttal improves the empirical breadth, reviewers with more critical stances remain unconvinced that the added evidence fully resolves concerns about contribution and underlying mechanisms. In particular, some reviewers consider the gains of the proposed method to be relatively small on strong baselines, and the mechanistic explanation of how alignment leads to cultural knowledge degradation remains largely correlational rather than causal (Reviewer NjxX; Reviewer cuHY).

Taken together, while the paper offers a useful framing and careful empirical analysis, the reviewers’ concerns regarding novelty, the strength of assumptions, and the depth of mechanistic insight remain substantial and ultimately inform my recommendation to reject.

**Reviewer Concerns:**

Several reviewer concerns were partially addressed by the rebuttal, while others remain outstanding.

Addressed or partially addressed concerns:
- Generalization across models and datasets.
  Multiple reviewers questioned whether the findings generalize beyond a single model and dataset (Reviewer cuHY; Reviewer uUYH; Reviewer 73cB). The authors addressed this by adding experiments on an additional model (Qwen2.5-14B) and an additional culturally specific dataset (CulturalBench), showing that the transfer–localization trade-off and the behavior of SUR-Steering persist. This directly responds to the concern about single-model evaluation and improves empirical coverage.

- Methodological clarity and reproducibility.
  Reviewers requested clearer explanations of the LOC-steering vector and configuration procedure (Reviewer uUYH; Reviewer 73cB). In the rebuttal, the authors provide a concrete description of how LOC-steering vectors are constructed from paired contextualized and decontextualized inputs, as well as how layers are selected using layer-wise analysis. This improves transparency and reproducibility.

- Whether cultural knowledge is erased or merely suppressed.
  Reviewer 73cB explicitly asked whether cross-lingual alignment permanently erases cultural knowledge or suppresses access to it. The authors conducted an additional explicit-context prompting experiment showing that reintroducing localization cues recovers cultural performance but degrades global transfer, supporting the interpretation that knowledge is suppressed rather than fully erased. This is a meaningful and well-aligned rebuttal to that concern.

Outstanding concerns:
- Novelty and contribution beyond existing work.
  Despite the clarifications, reviewers who were skeptical about novelty remain so. The rebuttal emphasizes reframing and the combination of existing steering vectors at different layers, but does not convincingly demonstrate that the proposed method constitutes a fundamentally new technique rather than a reasonable engineering combination of known approaches (Reviewer cuHY; Reviewer NjxX; partially Reviewer uUYH).

- Strength and scope of the language-as-culture-proxy assumption.
  Although the authors acknowledge this assumption and discuss its limitations in the rebuttal, it remains a central pillar of the analysis. Reviewers raised concerns that cultural knowledge often transcends language boundaries (e.g., region-specific knowledge within the same language), and the rebuttal primarily offers conceptual extensions rather than empirical validation. As a result, the generality of the conclusions remains limited (Reviewer 73cB; Reviewer NjxX).

- Depth of mechanistic explanation.
  While the authors provide representational analyses and quantitative similarity measures, some reviewers remain unconvinced that the paper offers a deep mechanistic understanding of how alignment objectives lead to cultural knowledge degradation, as opposed to descriptive or correlational evidence (Reviewer cuHY; Reviewer NjxX).

Overall, the rebuttal strengthens the empirical side of the paper but does not fully resolve concerns about contribution, assumptions, and explanatory depth.

**Reviewer Scores:**

Based on the rebuttal and discussion, I estimate the following likely score changes if reviewers had been able to fully participate:

- Reviewer cuHY (original overall rating: 2 – reject).
  While the additional experiments address the single-model concern, the rebuttal does not substantially change this reviewer’s position on novelty and contribution. I do not expect a meaningful score increase; the rating would likely remain a reject (2), or at most increase marginally to a low borderline score.

- Reviewer NjxX (original overall rating: 4 – marginally below acceptance).
  The added generalization experiments and clarifications partially address this reviewer’s concerns, particularly regarding scope and applicability. However, concerns about limited innovation and shallow mechanistic insight likely remain. I would expect a small upward adjustment at most, but still below the acceptance threshold.

- Reviewer uUYH (original overall rating: 6 – marginally above acceptance).
  This reviewer’s main concerns regarding generalization and technical clarity are largely addressed in the rebuttal. However, their reservations about novelty remain partially unresolved. I expect this reviewer would likely maintain a similar score (around 6), but not significantly increase it.

- Reviewer 73cB (original overall rating: 8 – accept).
  The rebuttal directly engages with many of this reviewer’s questions and strengthens the paper along the dimensions they care about. I would expect this reviewer to maintain their positive assessment (around 8), with little change.

Overall, the score distribution would likely remain polarized, with one strong accept, one marginal accept, and two reviewers remaining below the acceptance threshold, which supports the reject recommendation.

---

### Decision · Program_Chairs · 2026-01-26

Reject